



# Widespread slowdown in thinning rates of West Antarctic Ice Shelves

Fernando S. Paolo[1], Alex S. Gardner[1], Chad A. Greene[1], Johan N. Nilsson[1], Michael P. Schodlok[1], Nicole-Jeanne Schlegel[1], Helen A. Fricker[2]

[1] Jet Propulsion Laboratory, California Institute of Technology, Pasadena, CA

[2] Scripps Institution of Oceanography, University of California, San Diego, CA

*Correspondence to*: Fernando Paolo (fernando@globalfishingwatch.org), Alex Gardner (alex.s.gardner@jpl.nasa.gov)

**Abstract.** Antarctica's floating ice shelves modulate discharge of grounded ice into the ocean by providing backstress. Ice shelf thinning and grounding line retreat have reduced this backstress, driving rapid drawdown of key unstable

areas of the Antarctic Ice Sheet. If ice shelf loss continues, it may initiate irreversible glacier retreat through the marine ice sheet instability, leading to significant sea level rise. We analyze 26 years (1992—2017) of changes in satellite-derived Antarctic ice shelf thickness, flow and basal melt rates to construct a time-dependent dataset and investigate temporal variability. We found an overall pattern of thinning around Antarctica, with a thinning slowdown starting around 2008 widespread across the Amundsen, Bellingshausen and Wilkes sectors. We attribute this slowdown partly

to modulation in external ocean forcing, likely altered in West Antarctica by negative feedbacks between ice shelf thinning rates and grounded ice flow, and sub-ice-shelf cavity geometry and basal melting. Our satellite-derived ice-shelf thickness and basal melt dataset uses a novel data fusion approach, state-of-the-art satellite-derived velocities, and a new surface mass balance modeling. We test the resolution capability of these data with an ice-ocean modeling experiment.

**1 Introduction**

The Antarctic Ice Sheet is Earth's largest reservoir of freshwater, with global sea level equivalent of ~58 m (Morlighem et al., 2020). The rate at which Antarctica's ice is discharged to the ocean is controlled by its fringing ice shelves, which exert a resistive "buttressing" force on the upstream grounded ice. In recent decades, ice shelves in West Antarctica's Amundsen Sea Embayment (ASE) have rapidly thinned (Paolo et al., 2015) and in response, the

grounded glaciers that feed them have accelerated (Mouginot et al., 2014; Konrad et al., 2017; Rignot et al., 2019). There is some evidence that a similar drawdown and acceleration might also have started in some basins of East Antarctica (Roberts et al., 2018; Khazendar et al., 2013). This suggests that a reduction in buttressing may have already initiated a process of runaway retreat in regions that are inherently unstable due to the marine ice sheet instability (Weertman, 1974; Joughin et al., 2014; Rignot et al., 2014). Thus, over the coming century the mechanisms that

control ice shelf thickness change will be inextricably linked to global sea level, and improving our understanding of these processes will increase our ability to accurately predict Antarctica's overall contribution.

The processes that drive ice shelf thickness changes are directly linked to the atmosphere and ocean, and vary on different timescales (Paolo et al., 2018; Adusumilli et al., 2020). Ice shelves gain mass through the lateral influx of



ice across the grounding line (the boundary where glaciers become afloat), and local surface accumulation via
snowfall; mass is lost by calving, basal melt, and minor surface effects such as surface melting, wind scour,
sublimation, and surface runoff. Ice shelf thickness can be estimated from satellite radar and laser altimetry, and
previous studies have attributed observations of ice shelf thinning to increased rates of basal melting (Pritchard et al.,
2012; Adusumilli et al., 2020); however, direct measurements of basal melt rates are scarce (Jacobs et al., 2013;
Jenkins et al., 2018; Christianson et al., 2016; Dutrieux et al., 2014). Basal melt rates can be inferred from satellite-
derived thickness field combined with other inputs (ice divergence and surface mass balance), and previous large-
scale melt-rate estimates made using this technique have considered mean rates for periods of less than a decade
(Depoorter et al., 2013; Rignot et al., 2013), or have been limited in spatial resolution over longer time periods
(Adusumilli et al., 2020). These data limitations have hindered our ability to identify the driving mechanisms and how
they vary with time. To better quantify spatial and temporal changes in both ice thickness and basal melt rates, we
combined observations from four satellite radar altimeters to derive a pan-Antarctic time series of ice shelf thickness
and basal melt rates with the ability to resolve kilometer-scale structures (3-km grid at 3-month intervals) consistently
from 1992 to 2017. By correcting for time-variable ice divergence in critical locations where significant ice thinning
and flow acceleration have occurred, we are able to disentangle the oceanic, atmospheric and dynamic components of
recent ice shelf change, and show how their rates vary with time.

## 2 Data and Methods

### 2.1 Radar altimetry data

We used data from four European Space Agency (ESA) satellite radar altimetry missions: ERS-1 (1991-1996), ERS-
2 (1995-2003), Envisat (2002-2010) and CryoSat-2 (2010-2017). The first three satellites carried conventional pulse-
limited altimeter systems with a footprint size of less than 3000 m in diameter over flat areas, sampling along flights
55  every ~370 m, with a latitudinal coverage up to 81.5° S. CryoSat-2 (currently in operation) carries a dual antenna
Doppler/delay altimeter, operating in synthetic aperture radar interferometric (SARIn) mode over the margins of the
ice sheet including the ice shelves, with a latitudinal coverage up to 88° S. This altimeter yields along-track and across-
track footprint sizes of about 300 m and 3000 m, respectively, sampling along flights every ~370 m. ERS-1 and ERS-
2 operated in different modes of acquisition over the Antarctic ice shelves: 'ocean mode' (330 MHz) with a finer
60  sampling of the return radar echo, which translates to a higher-resolution radar waveform; and 'ice mode' with a
coarser sampling of the waveform (82.5 MHz), mostly to keep track of radar echoes interacting with terrain
undulations on the ice sheet margins and interior.

We obtained ERS-1 and ERS-2 data from the "REprocessing of Altimeter Products for ERS (GDR): 1991 to 2003"
(REAPER) (Brockley et al., 2017), as the product provides updated corrections and improved calibrations. We
65  obtained Envisat data from the "RA-2 Geophysical Data Record" (GDR) v2 (http://www.uat.esaportal.eu/web/guest).
We obtained CryosSat-2 data from "ESA L1b Baseline-C product" (Bouffard et al., 2018), using our own CryoSat-2
processor (Nilsson et al., 2016).

### 2.2 Data processing



We derived surface heights for each mission using the standard 30% threshold retracker (ICE-1) available for all products, except for CryoSat-2 where we used our in-house-developed retracker for the SARIn mode (Nilsson et al., 2016). The largest source of uncertainty in radar altimetry is the slope-induced error (Brenner et al., 1983), which depends on the magnitude of the surface slope and can range from 0 to 100 m (0 from 1 deg). We minimized this error by applying a slope correction based on the Digital Elevation Model Bedmap2 (Fretwell et al., 2013).

To measure the changes in ice-sheet surface height the topography must be removed. We used a method similar to Nilsson et al. (2016), McMillan et al. (2014), and Wouters et al. (2015), but with some fundamental modifications. In previous studies, the time-varying and static topography have been solved for in one least-squares inversion. This approach has an inherent limitation as generating time series of high temporal sampling requires a search radius of 1-3 km, which does not usually allow for good estimation of the underlying topography that requires smaller spatial scales (< 1 km). To this end, we first separated the data into ascending and descending orbits (as well as ice and ocean modes) to mitigate the impact of any inherent biases. We also filtered each ground track with a 3-point moving median to remove potential anomalous height measurements, e.g. from off pointing. We then solved for the static topography independently on each data set using a biquadratic or bilinear surface model (depending on the number of available data points), which is removed to obtain the time-varying height change signal. This approach allows us to accommodate the different correlation lengths of the processes affecting the retrieval of height-change time series, by permitting the use of independent search radii for the different steps included in the data processing workflow.

Another major limitation of radar altimetry measurements is the variable penetration depth of the radar signal due to changes in the scattering by snow/firn surfaces. This effect usually introduces artificial trends and inflates seasonal amplitudes of radar-derived time series. Previous studies (Sørensen et al., 2018; Wouters et al., 2015; Adusumilli et al., 2020) mitigated this effect by including a backscatter term in the full inversion step, together with the topography and height change terms. Instead, we performed this operation independently, optimizing the search radius to maximize the temporal coverage of each time series and the correlations to the different radar waveform parameters.

We set the inversion cells (i.e. search centroid and radii) following clusters of repeat tracks (along-track processing), leaving the gridding procedure for a later stage where we take into account the optimal spatial and temporal scales of each estimated quantity; a key difference with previous studies (Paolo et al., 2015; Adusumilli et al., 2020). Our inversion cell sizes vary linearly with latitude to account for data density, decreasing with latitude as satellite ground track spacing becomes denser.

Our ice shelf boundary definition is derived from a combination of Landsat imagery and ICESat data (Depoorter et al., 2013), updated for later epochs with the Measures v2 boundaries (Rignot et al., 2017) and manually edited for significant calving events from satellite imagery. To avoid ice shelf data contamination with grounded ice data due to inaccuracies in the boundary determination and to mitigate unaccounted variability in the ice-shelf front position, we removed (i) a 3-km buffer around the ice shelf perimeters from our data set, this is based on the pulse-limited footprint (~3 km) of standard radar altimeters over flat areas, and (ii) an additional 3-km buffer (6 km total) from the ice-shelf fronts for the analysis presented here to avoid errors resulting from changes in ice shelf area through time not captured in our data.



### 2.3 Geophysical corrections

Various geophysical corrections are required for radar altimetry elevation data over ice shelves (Paolo et al., 2016). We describe the specific corrections we applied below:

*Standard Correction*s: We applied the dry troposphere, wet troposphere, ionosphere, solid earth tide and pole tide corrections that are provided with the data.

*Surface Slope*: Because the radar altimeter echo is sensitive to terrain slope, reflecting off the Point of Closest Approach (POCA) to the satellite, we relocated the observations to the POCA and corrected the range using the relocation method described in (Bamber et al., 1994), which is based on surface slope, aspect and curvature information estimated from the Bedmap2 DEM (Fretwell et al., 2013).

*Inverse Barometer*: We calculated the inverse barometer correction using ERA-Interim mean sea level pressure (Dee et al., 2011). Instead of the standard calculation that uses the global mean pressure over the ocean as the "reference pressure" (Le Traon et al., 1998), we used the climatological mean at each grid-cell location as our reference value, providing a spatially-varying reference pressure field.

*Ocean Tides*: Ice shelves respond instantaneously to the rising and falling ocean tide, so this signal needs to be removed from the measured heights. We removed the original tide corrections that were applied to the raw height data and instead applied improved tides using the regional Circum-Antarctic Tidal Simulation model (CATS2008a), an updated version of the inverse tide model described by (Padman et al., 2002). This model has a higher resolution (~4 km) and a more accurate land mask than global models, resulting in more accurate tide prediction close to the coast. We also corrected for the ocean tidal loading (the elastic deformation of the seabed in response to the ocean-tide load) using the TPXO7.2 model (Egbert and Erofeeva, 2002).

*Mean Sea Level and Trend*: Previous studies have estimated mean sea level from low-order (i.e. low resolution) geoid models and global Mean Dynamic Topography (MDT) fields. Low-order geoid models contain substantial artifacts at the ocean-ice-land transition (Armitage et al., 2018), which introduces large biases to the estimated mean sea level underneath the ice shelves. Global MDT fields are extrapolated from the distant edge of the sea ice to the ice-shelf grounding lines, simply because these global data sets do not contain measurements within the area covered by sea ice, propagating incorrect ocean-state features underneath the ice shelves. Instead, we used Mean Sea Level directly measured with altimetry all the way to the ice-shelf fronts through the sea-ice leads at 3 km resolution (Armitage et al., 2018). We then removed the high-resolution Combined Gravity Model GOCO05c (Fecher et al., 2017) to obtain the MDT field. This geoid model combines modern gravity missions like GRACE and GOCE with in situ observations to provide unprecedented detail of the gravity field underneath the ice shelves. We extended the residual MDT with a Gaussian average tapering to zero at the grounding lines, and then added back to the geoid. We also extended the sea level trend field (from CNES/AVISO) in the same way.

*Surface Scattering*: The radar altimeter return waveform shape is governed by the degree of surface scattering and volume scattering (Davis and Moore, 1993; Partington et al., 1989). Over the years many studies have used an empirical correction based on removing the correlation between changes in the observed height and changes in the



shape of the radar waveform (Nilsson et al., 2016; Wingham et al., 2009; Zwally et al., 2005; Davis and Ferguson, 2004; Paolo et al., 2016). Here, we used estimates of the radar backscatter coefficient, leading edge width and trailing edge slope (Rémy and Parouty, 2009; Khvorostovsky, 2012) to characterize changes in the shape of the radar waveform (Figure 1). We normalized these parameters by their standard deviation and then regressed them, by means of robust multivariate regression, against the residual time series of height changes (at the point-measurement level)

to determine a linear combination of sensitivity gradients, which we then used to remove temporally changing scattering effects from the original time series. To maximize correlations and reduce erroneous correlations to geophysical trends, we applied a difference operator to each parameter time series before the regression. (Figure 1)

**2.4  Firn and surface mass balance modeling**

Measured height changes reflect, in addition to ice change, the changes in air content within the firn layer. To model

the evolution of firn density (i.e. total column of firn air content: FAC) and Surface Mass Balance (SMB), we utilized the Glacier Energy and Mass Balance model (GEMB: run under development r24739). GEMB is a module of NASA's open-source Ice Sheet and Sea Level System Model (ISSM). It is a column model (no horizontal communication) of intermediate complexity, which includes those processes deemed most relevant to glacier studies but also retains computational efficiency that can accommodate the very long (thousands of years) spin-ups necessary for initializing

deep firn columns. The model simulates parameterized snow grain growth, depth-dependent albedo based on grain-size (Brun et al., 1992; Gardner and Sharp, 2010), thermal diffusion, depth-dependent calculation of temperature, melt, meltwater percolation and refreeze, parameterized snow grain growth, compaction of snow and firn down to required depths for full compaction to glacial ice, calculation of turbulent and evaporative fluxes, and dynamic management of firn-layer thickness (Figure 2).

GEMB calculates the energy and mass balance of a column of snow and firn at 3-hourly intervals. Inputs are 2-meter surface temperature, 10-m wind speed, downward shortwave and longwave radiation fluxes, precipitation, surface pressure, and screen-level vapor pressure. Here, we used 3-hourly inputs from the ERA5 reanalysis (C3S, 2017).

At every 3-hourly time step, the model first calculates the evolution of the snow grain size, including the effective grain radius, grain dendricity, and grain sphericity, accounting for dendritic (Brun et al., 1992), non-dendritic

(Marbouty, 1980), and wet snow metamorphism (Brun, 1989). Next, the albedo module calculates the modeled snow, firn, and ice albedo based on the model estimates of effective grain radius, through the summation of albedo within spectral bands within a prescribed solar light distribution (Lefebre et al., 2003) or as a function of specific surface area (Gardner and Sharp, 2010), based on user preference. Here, the albedo calculation follows that defined in (Lefebre et al., 2003). Depending on the calculated albedo, a percentage of the shortwave radiation downward is absorbed at the

surface. Shortwave radiation then penetrates the surface and is absorbed through the underlying snow to depth (Vionnet et al., 2012).

GEMB next calculates the evolution of the temperature of the snowpack through thermal diffusion (Sturm et al., 1997), in response to the forcing of surface air temperature, radiative fluxes, and sensible and latent heat fluxes. For stability, diffusion is calculated at time steps finer than 3-hourly, as an even interval of seconds in an hour, and the thermal

model loops at the finer time step, calculating the diffusion of temperature downward through the layers of the



snowpack. Surface radiative absorption and emission fluxes are determined, and then sensible and latent heat fluxes (Murphy and Koop, 2005), rates of evaporation/condensation, and a new temperature profile are determined at each diffusion time step.

GEMB then manages accumulation, or the addition of mass to the top of the snowpack, determines the density of the
accumulated mass, and manages the properties of the top layer to account for the newly accumulated mass. If the air temperature is less than the melting point, the surface snow accumulates within the top layer of the snowpack. Here, surface snow density is calculated after (Kaspers et al., 2004), and the initial snow dendricity, sphericity, and grain radius are set after (Vionnet et al., 2012). If the air temperature is above or at the melting point, then accumulation is added to the system as liquid water (i.e. rain). In this case, the temperature and density of the surface layer is adjusted
to account for the refreeze of the added mass as ice. For these simulations, the top 10 m of the snowpack is restricted to a maximum layer thickness of 5 cm, with a minimum thickness of 2.5 cm. If the depth of the added layer in any 3-hour time step exceeds 2.5 cm, then it is added as a new top layer of snow, and if it is less than 2.5 cm it is combined with the existing top layer through adjustment of layer variables as a linearly weighted function of mass. The temperature of new snow is assumed to be the same temperature as the air.

Following the accumulation, GEMB determines how much melt will occur throughout the snowpack, how much will percolate into layers at depth, how much will refreeze, and how the snow temperature will change in response to these processes. First, if any amount of pore water that exists within the firn layer can be refrozen without heating the snow above the freezing point, it freezes, and the physical and thermal state of each model layer is updated accordingly. Next, starting at the surface, if the local thermal energy within the layer exceeds energy to melt the entire cell, the cell
is completely melted, and any temperature and melt surplus is redistributed to the layer beneath. If at any point, the meltwater reaches an impermeable layer within the snowpack, that water is considered runoff. If the pore space of the lower layer can accommodate the incoming water, it is retained and added to any local melt. If the total melt exceeds the irreducible water content (i.e. water held in place by capillary forces), it percolates into the layer below. Any remaining water is refrozen until the freezing point temperature is reached. Any melt that exits the deepest layer is
also considered runoff. The melt module also manages the properties of the layers, determining when to merge cells and when to split cells, and updating the snowpack properties accordingly. Additionally, the module ensures that the model layering adheres to the user-defined minimum layer thickness (here set to 2.5 cm), the max allowable rate of thickness change between each consecutive layer (here set to 10%), and the maximum total model depth (here set to 250 m). Before returning, this function checks that mass and energy were conserved, and if not, it throws an error.

Once melt is calculated and the layers are managed, GEMB determines the compaction of snow and firn. For all methods of calculating the densification, firn with density <=550 kg/m³ and firn with density >550 kg/m³ are treated independently. At these two different profile horizons, rate parameters (c0 and c1, respectively) are determined depending on which of the seven available densification method is being executed. These rate parameters are used to calculate densification rates and a new density profile, after (Herron and Langway, 1980).

After determination, GEMB makes a final calculation of the sensible and latent heat fluxes as well as evaporation/condensation from the snowpack surface, with consideration to the new surface properties. The bulk





Richardson number is calculated to determine the stability factors for weighting the sensible and latent heat fluxes (Ohmura, 1982), and then the fluxes are calculated based on 2 meter air temperature, surface temperature, and surface pressure. For the latent heat flux, the surface vapor pressure is calculated for ice or liquid water (Murphy and Koop,
2005) dependent on whether or not the snowpack is, respectively, dry or melting at the surface. Finally, evaporative mass change is calculated based on the latent heat flux. SMB is determined as accumulation plus the mass change due to sublimation/evaporation/condensation processes, minus runoff.

To produce the FAC and SMB time series output used for this study, we calibrated the snow-densification parameters to improve the agreement between model density profiles and observations. Here, the snow densification rate
parameters rely on the semi-empirical model for dry snow densification described in Appendix B of (Arthern et al., 2010), where the c0 and c1 parameters are corrected by parameters trained through comparison between modeled and measured ice core density profiles, after (Ligtenberg et al., 2011). That is, the calibration consists of calculating the model-to-observed depth of density horizon 550 kg/m$^3$ (MO$_{550}$) and the model-to-observed depth of density horizon 830 kg/m$^3$ minus depth of density horizon 550 kg/m$^3$ (MO$_{830}$) for all ice core locations.

A total of 78 Antarctic shallow cores are used for the training of c0 (from the surface to a density of 550 kg/m$^3$) and 25 deep Antarctic cores are used for the training of c1 (for densities >550 kg/m$^3$) (Smith et al., 2020; Montgomery et al., 2018). For the model values, GEMB is forced with a repeated cycle of 3-hourly ERA5 reanalysis from 1979 through 2009, for 250 cycles or 7750 years. For this model-calibration simulation, snow density is calculated using the uncalibrated (Arthern et al., 2010) densification expressions. The modeled horizon depths are derived using the
depth-density profiles at the end of the simulation at the model elements closest to core sites.

For all ice core locations, MO$_{550}$ and MO$_{830}$ values are plotted independently against the natural log of the local annual average accumulation rate ($C$) from 1979 through 1999, and a line is fit for each. The resulting linear expressions represent how the model-to-observed depths vary as a function C. The calibration results in the following expressions for MO$_{550}$ and MO$_{830}$:

$$\text{MO}_{550} = 2.838 - [0.316 \cdot \ln(C)], \qquad r^2 = 0.206$$
$$\text{MO}_{830} = 3.100 - [0.371 \cdot \ln(C)], \qquad r^2 = 0.643$$

GEMB is then run using the derived MO$_{550}$ and MO$_{830}$ expressions to correct densification rate. That is, the (Arthern et al., 2010) densification expressions (c0 and c1) are multiplied by the derived accumulation-dependent depth-density calibration parameters, MO$_{550}$ and MO$_{830}$, respectively (Ligtenberg et al., 2011). Note that, a minimum value of 0.25
is also imposed on MO$_{550}$ and MO$_{830}$, following (Ligtenberg et al., 2011).

Using the calibrated densification scheme, GEMB is run over the entire Antarctic Ice Sheet, at a spatial resolution ranging from 9 km near the margins of the ice sheet and areas of high gradients in SMB and ice velocity. The model decreases to roughly 50 km in the interior plateau of the ice sheet. The model is run for 7750 years (repeating 3-hourly ERA5 forcing from 1979 through 2009), to build up a layer of firn layer and spin-up firn properties. Following this
relaxation, the model is then forced with 3-hourly ERA5 reanalysis from 1979-2017, resulting in daily estimates of FAC and SMB. Results are linearly interpolated onto a constant 5-km grid for ice-shelf melt rate analysis, for comparison against other products, and the estimation of model uncertainties.





Uncertainties in the GEMB FAC and SMB records are estimated by comparison to other firn models (GSFC FDMv1 (Smith et al., 2020; Medley et al., 2020) and IMAU FDM – RACMOv2.3 (Ligtenberg et al., 2014; Medley et al., 2020)) at the timescales of our melt rate estimation (Figure 3). To do this we first interpolated all models onto the same 5 km grid. At each grid node, we constructed 5-month intervals of 5-daily (FAC) and monthly (SMB) rates. We then calculated the standard deviation of those 5-month intervals and respective degrees of freedom (independent estimates), as a measure of model dispersion at different epochs (more on errors below) (Figure 4).

**2.5 Data fusion**

We fused data from multiple satellites with different error characteristics and spatial distribution using an Optimal Interpolation approach (a.k.a. Gaussian Processes). We used four key metrics to produce continuous fields at 3-km posting every 3 months: distance between observations, distance of observations to grid nodes, observation errors, and along-track long-wavelength correlated errors [28] that we estimated empirically so as to minimize the variance of the interpolated field. To do this, we first aggregated data from each individual mission in 5-month bins for every 3-month time step. We calculated empirical covariances as a function of data separation at random locations over the ice shelves, and fitted analytical covariance models that we used to guide the interpolation (Melnichenko et al., 2014). The covariances describing the characteristic correlation lengths are computed for each parameter (Figure 5). We also used a latitude-dependent search radius to account for increased data density towards the pole as the satellite tracks get closer to each other (Figure 6).

At each grid cell, we then filtered and interpolated time series residuals larger than 5 standard deviations from the trend (defined by a piecewise polynomial fit). We also cross-calibrated the gridded records from the four satellite missions by computing the offsets (median of differences) between a low-pass-filtered version of the time series during the periods where consecutive missions overlapped. We filtered these records with a 5-point moving average (~1.25 years) to remove the effect of seasonality.

**2.6 Thickness change and basal melt rate inversion**

We performed our melt estimation on a Eulerian reference frame. There are two fundamental steps that we improved to estimate ice shelf basal melting from measured surface height. First, inverting height to thickness:

$$H(t) = \frac{\rho_w}{\rho_w - \rho_i}(h_0 + \Delta h_{altim} + \Delta h_{tide} + \Delta h_{load} + \Delta h_{IBE} + \Delta h_{FAC} + \Delta h_{MSL} + \Delta h_{SLT})$$

where $H$ is thickness, $\rho_w$ and $\rho_i$ are density of ocean water (1028 kg/m³) and ice (917 kg/m³), respectively, $h_0$ is mean surface topography from CryoSat-2 referenced to 2014, $\Delta h_{altim}$ is height change from altimetry, and the respective corrections for tides, load tide, inverse barometer effect (IBE), firn air content (FAC), mean sea level (MSL), and regional sea-level trends (SLT). Second, solving the mass balance equation:

$$\dot{b}(t) = \frac{\partial H}{\partial t} + \nabla \cdot (H\boldsymbol{u}) - \dot{a}$$

where $\dot{b}$ is total basal melt rate, $\partial H/\partial t$ is thickness change rate, $\nabla \cdot (H\boldsymbol{u})$ is ice-flux divergence, with $H = H_0 + H(t)$ and $\boldsymbol{u}$ being velocity, and $\dot{a}$ is surface accumulation rate (SMB). We note that all the terms in this equation are time





dependent, with velocity varying in time for the Amundsen Sea sector (Mouginot et al., 2014; Gardner et al., 2018) only and assumed constant outside this region due to lack of data. This assumption is justified by the relatively small flow changes observed outside the Amundsen Sea sector in the past couple of decades (Gardner et al., 2018; Rignot et al., 2019). We computed all of our spatial and temporal derivatives implicitly, using a piecewise overlapping

polynomial fit (Savitzky and Golay, 1964) that was designed to handle noisy data. We used a yearly fit window for temporal derivatives, and a 15 x 15 km fit window for spatial derivatives.

InSAR-derived velocities contain substantial artifacts from ionospheric effects and residual tidal displacements. These artifacts are amplified by the spatial derivatives and map onto the melt-rate estimates. We constructed a velocity product for the Amundsen Sea and Bellingshausen Sea sectors by combining data from (Mouginot et al., 2014)

[1996—2012] and (Gardner et al., 2018) [1985—2018] with complete coverage only after 2014). The first year with significant coverage is 1996 when InSAR ERS-1 data were collected over the ice sheet. Velocities are first mapped to the same 240-m grid as used by (Gardner et al., 2018). Velocity estimates falling outside of mapped ice extents that are updated at the end of 2012 and 2013 are set to no data values. We then define the reference velocity as the 1996 velocity field or the earliest valid measurement thereafter. For areas moving faster than 200 m/yr we calculate percent

anomalies relative to the reference velocity for each product separately, filter with a 5-km windowed moving median, then merge products by taking the mean of each year. Years with less than 30% coverage for fast moving ice (>= 200 m/yr) are discarded. If missing annual values are within 25 km of a valid datapoint they are filled using natural neighbor interpolation, otherwise anomalies are set to zero. Outside of fast-moving areas, annual anomalies are tapered to zero using a 10-km cosine taper. Annual anomalies are smoothed one last time using a 5-km windowed moving

mean. To create a continuous record of velocity, annual anomalies are interpolated in time using a spline interpolant and multiplied by the reference velocity. We then smoothed the 240-m resolution velocity fields with a Gaussian kernel with standard deviation of 1.2 km prior to re-gridding onto our 3 km final grid (Figure 7).

Since the southern limit of ERS-1, ERS-2 and Envisat was 81.5° S, we have no data between 81.5° S to 88° S prior to CryoSat-2 (2010). To overcome this, we made the assumption that mean thickness and basal melt rates were constant

prior to 2010. This assumption is based on the insignificant changes in thickness and melting observed over those regions during the CryoSat-2 era.

We estimated acceleration/deceleration in ice shelf thinning from the trend in the thickness-change rate time series. We fitted nonlinear trends using the Savitzky-Golay filter (Savitzky and Golay, 1964), which unlike an ordinary least squares line fit, is robust to outliers and sudden changes at the beginning and end of the records. Mean acceleration

over the timespan of our records is then defined as

$$\bar{a} = \frac{\Delta \widetilde{\partial H}}{\Delta t}$$

where $\widetilde{\partial H}$ is the mean rate of thickness change from both ends of the trend fit, with $\Delta t = 26$ years. Note that this is a robust estimate of acceleration (or second derivative), compared to the slope of a straight-line fit, as sudden changes in the trend are captured by our nonlinear trend fit. Finally, we compute the thickness change, basal melt, divergence,

and SMB 2D mean fields from the mean of the respective instantaneous rate time series, as



$$\overline{\partial H} = \frac{1}{n} \sum_k \partial H_k$$

where $\partial H_k$ is the time-evolving rate of thickness change and $n$ is the number of samples in each grid cell record.

### 2.7 Ice-ocean modeling

We simulated ice-ocean interactions using the Massachusetts Institute of Technology general circulation model
(MITgcm) that includes a dynamic/thermodynamic sea-ice model (Losch et al., 2010). Freezing/Melting processes in
the sub-ice-shelf cavity are represented by the three-equation-thermodynamics of (Hellmer and Olbers, 1989) with
modifications by (Jenkins et al., 2001) as implemented in MITgcm by (Losch, 2008).

The model domain (Figure 8) is derived from the global configuration (LLC1080) used by the Estimating the
Circulation and Climate of the Ocean (ECCO) project (Forget et al., 2015), with a nominal horizontal grid spacing of
~3 km on the Antarctic continental shelf, comprising the region of the Amundsen Sea used in (Schodlok et al., 2012).
The vertical discretization is enhanced, compared with that used by the ECCO project, to 113 vertical levels of varying
thickness in order to capture the deep part of the sub-ice-shelf cavities near the grounding line. The bathymetry is a
blend of IBCSO (Arndt et al., 2013) outside the sub-ice-shelf cavities and BedMachine Antarctica elsewhere
(Morlighem et al., 2020). We derived ice shelf draft, the key component of our modeling experiment, from our ice
shelf thickness data. Initial conditions and boundary conditions for hydrography (*T, S, u, v*) and sea ice are derived
from a coarse resolution global state estimate (~20 km horizontal grid spacing, LLC270) for the integration period
1992 to 2017. Due to the difference in resolution between the global integration and the ~3-km model domain, a
relaxation is applied to temperature and salinity at the boundaries (10 grid points into the model domain) to avoid
artifacts such as wave energy radiating into the model interior, and 5 grid points for sea ice variables. Surface forcing
335    is also provided by the ECCO project. We have successfully applied similar configurations to study ice-shelf/ocean
interactions on the cube sphere as well as the LLC grids (Khazendar et al., 2013; Nakayama et al., 2017, 2018).

In our simulations we investigate the sensitivity of ice shelf basal melt to changes in ice shelf draft, hydrographic
properties and surface forcing. The model configuration comprise the ice shelf drafts and hydrography of the years
1993 and 2017. The year 1993 represents a year with a deeper draft and colder ocean properties on the continental
340    shelf, while the year 2017 represents a year with shallower draft and warmer ocean properties. The initial integration
starts from the LLC270 hydrography in 1992 (2016 respectively) for one year as a spin up. After the spin up, the
surface forcing of the cold year 1993 (warm year 2017) is used as a repetitive forcing for 10 years of integration. The
output of the last year of integration is averaged and its results analyzed. Additionally, we used the ice shelf draft of
1993 (2017) in the warm (cold) hydrography and surface forcing, and integrated for 10 years with the last year being
averaged and analyzed. Thus, we have 4 model simulations: Deep & Cold, Shallow & Cold, Deep & Warm, Shallow
& Warm (Table 1).

### 2.8 Uncertainty quantification

Since in situ basal melt rate measurements are not available at large scale for Antarctica, and only a few localized (in
time and space) indirect estimates exist, we provide a formal statistical error by identifying and propagating the first-
order uncertainties affecting our satellite estimates of ice shelf basal melt.





For each grid cell, we estimated the 26-year variance of the height records, which is a conservative measure of error as it accounts for both random fluctuations and geophysical signals. This error therefore reflects the complexity of local topography and our ability to model it, any residual tide and backscatter variability not fully accounted for, cross-calibration errors, and inherent variability such as seasonality. We then have

$$e_h = \frac{1}{\sqrt{n}} std(h - h_{trend})$$

$$e_H = \frac{\rho_w}{\rho_w - \rho_i} \sqrt{e_h^2 + e_{FAC}^2}$$

where $h$ is the height time series comprised of six independent data sets (4 missions and 2 modes of operation), so we set $n = 6$, $h_{trend}$ is the trend from a piecewise polynomial fit, $e_H$ is thickness error, $e_{FAC}$ is modeled firn air content error, $\rho_w$ and $\rho_i$ are the densities of ocean water (1028 kg/m³) and solid ice (917 kg/m³), respectively.

Assuming no significant error (relative to other variables) in the timestamps, the spatial coordinates, and the 26-year mean thickness, and assuming the same error in both velocity components ($u$ and $v$) of 5 m/yr (see below), we then have

$$e_{\partial H} = \frac{1}{dt} \sqrt{e_{H_k}^2 + e_{H_{k+1}}^2}$$

$$e_{\nabla} = \frac{1}{dx} 2H e_u$$

$$e_{melt} = \sqrt{e_{\partial H}^2 + e_{\nabla}^2 + e_{SMB}^2}$$

where $e_{\partial H}$ is the error in the thickness change rate, $e_{\nabla}$ is the error in the ice flux divergence, $e_u$ is the error in the velocity fields, $e_{melt}$ is the error in the basal melt rate, $e_{SMB}$ is the error in the modeled surface mass balance, with $dt = k = 1$ year (the time window used to estimate derivatives with a piecewise fit) and $dx = 3$ km (the grid spacing).


For the mean rate of change fields, we take the variance of the respective instantaneous rate-of-change records. Here, again, our error estimate is conservative as we are including natural variability as well as systematic trend changes, in addition to random errors, to the overall uncertainty. We then have the following error for the thickness change mean field

$$e_{\overline{\partial H}} = \frac{1}{\sqrt{n/s}} std(\partial H / \partial t)$$

where $\partial H / \partial t$ is the rate of thickness change time series, $n$ is the number of samples, and $s$ is a scaling factor to adjust the degrees of freedom for spatial correlation: $s = L^2 / A$, with $L$ being the spatial scale of the variable in question and $A$ our grid-cell area (~9 km²). The same expression is used for divergence and SMB. We set $L = 3$ km for thickness change rate (see Figure 5 for typical decorrelation lengths over the ice shelves), and $L = 31$ km for the smoother





divergence and SMB fields (which corresponds to the cell size of the ERA5 reanalysis grid). The mean basal melt
error is then obtained from a quadratic sum of these three errors.

Given that the acceleration term is derived from the end points of a trend fit to the instantaneous rate of change in
thickness, its error can be estimated by

$$e_{\overline{accel}} = \frac{\sqrt{2}}{dt}\, e_{\overline{\partial H}}$$

Then the errors for the ice-shelf average values are simply the aggregate of the errors over each ice shelf (for thickness
change, divergence, SMB and acceleration)

$$\langle e_{\partial H} \rangle = \frac{1}{n} \sum_i e_{\overline{\partial H}_i}$$

with

$$\langle e_{melt} \rangle = \sqrt{\langle e_{\partial H} \rangle^2 + \langle e_\nabla \rangle^2 + \langle e_{SMB} \rangle^2}$$

**2.9  Quality assessment**

To assess the reliability of our estimates we conducted a separate ice-shelf mass budget calculation, and compared our
results with published estimates using a control volume approach:

$$\frac{\partial M}{\partial t} = \overline{Q}_{in} - \overline{Q}_{out} + \overline{a} - \overline{b}$$

where $\partial M / \partial t$ is change in ice shelf mass, $\overline{Q}_{in}$ and $\overline{Q}_{out}$ are time-averaged (2010—2017) rates of ice fluxes across
the grounding line and ice front (calving), respectively, $\overline{a}$ is the time-averaged rate of surface accumulation (SMB), $\overline{b}$
is the time averaged basal melt rate for the period 2010—2017, and we compare our results to similar published data,
using the 500 m composite melt rate map from (Adusumilli et al., 2020) for the period 2010—2018.

To calculate ice shelf mass balances, we define a control volume for each ice shelf, starting with the ice shelf outlines
provided by (Mouginot et al., 2017b). We adjusted the control volume outlines to remove any grid cells that are
reported to have zero ice thickness data in BedMachine Version 2, are missing surface velocity data in the ITS_LIVE
mosaic, or do not have valid melt data in our estimates of mean melt rate for the 2010—2017 period. We then buffered
the remaining outlines inward by 3 km on all sides of each ice shelf to avoid uncertain ice thickness estimates near
grounding lines and to ensure confidence in ice flux interpolation near dynamic calving fronts. Furthermore, we avoid
complications that could be introduced by poorly understood effects of how bridging stresses might affect how basal
melt anomalies get transmitted as expressions of surface elevation change. With this in mind, we considered only grid
cells that are near hydrostatic equilibrium, which we identified using a simple ice flexure model with a threshold
defined by bedmachine_interp('flex', x, y) > 0.9 from (Greene et al., 2017). If multiple, unconnected sections of control
volume result for any ice shelf after adjusting the outlines, we use only the largest section for each ice shelf. As a final
step, we interpolated to a consistent 100-m spacing along the outline of the control volume. An example control
volume is shown in Figure 9.



We note that by our approach, the area defining each control volume may significantly underrepresent the full extent of each ice shelf, but our analysis is fully self-consistent and we do not directly compare our mass balance estimates to previously published estimates that have been calculated for different control volumes.

To calculate ice flux into and out of each control volume, we interpolated surface velocity across the control volume flux gate using itslive_interp('across', x, y) in Matlab (Greene et al., 2017), but where any missing data were filled with velocity estimates from (Rignot et al., 2017). Ice flux at each point along the control volume is calculated as the product of ice thickness and ice velocity across the flux gate. For ice thickness we used our mean measurement of ice thickness for the study period. We also repeated the analysis using ice thickness from BedMachine version 2, but found no significant difference. The total $\overline{Q}_{in} - \overline{Q}_{out}$ mass balance for each ice shelf is calculated as the sum of the ice fluxes measured along the control volume outline, multiplied by the 100-m spacing of our flux gate, and multiplied by ice density (917 kg/m$^3$) to convert volume to mass.

To account for ice mass gained or lost at the surface ($\overline{a}$), we converted GEMB mass balance outputs into units of Gt/yr per grid cell, then calculated the sum of the SMB values for all grid cells within the control volume.

For our $\overline{b}$, we generated a mean melt rate map for the years 2010—2017 (the CryoSat-2 period), resampled it to 500-m resolution, then converted the map into units of Gt/yr for each grid cell. The basal mass balance for every ice shelf is then obtained as the simple sum of the $\overline{b}$ grid cells within the control volume outline of each ice shelf. For comparison to previous studies, we follow the same procedure to calculate total basal melt for each ice shelf using Adusumilli et al.'s 500-m resolution composite melt rate map that spans 2010—2018. We used the provided w_b_interp field to fill in the missing grid cells, which they obtained by assigning melt rate based on a melt-depth relationship for each ice shelf.

Figure 10 shows basal melt rate estimates from our data and from the data published by (Adusumilli et al., 2020), compared to the basal melt rates that would be expected from two different control volume approaches. Panels (a) and (b) assume that the volume of each ice shelf remained constant over the observation period, meaning $\partial M/\partial t = 0$. In reality, most ice shelves experienced some change in thickness, which we account for in panels (c) and (d) by summing up the grid cells of a map of the linear trend in ice thickness that we generated from our data. In the flux gate calculation, we do not account for changes in velocity, but nonetheless, panel (d) shows good agreement between our estimated melt rates and the melt rates that would be expected from the control volume approach.

## 3 Results and Discussion

Our pan-Antarctic analysis reveals spatial patterns of ice-shelf thickness change (Figure 11) that show the signature of ocean-induced melting, with the highest thinning rates found adjacent to the deep grounding lines (Figure 12), where dense and warm modified Circumpolar Deep Water (mCDW) is more likely to be present and the pressure-melting point of ice is depressed. This overall pattern of ice shelf loss is consistent with previous estimates of ice shelf thinning (Paolo et al., 2015; Adusumilli et al., 2020; Shepherd et al., 2018); ice shelves in the ASE and Bellingshausen Sea Embayment (BSE) sectors, where the highest losses from the grounded ice are occurring, have all thinned over the past quarter century (Figure 13 and Figure 14). Our estimates for average ice shelf thinning rates in the ASE and





BSE over the 1992–2017 observational period is $2.6 \pm 0.3$ and $0.5 \pm 0.2$ m yr$^{-1}$, respectively, suggesting that these ice shelves are largely responding to a change in ocean conditions that predates our satellite altimetry record, with shorter term variability only resulting in small deviations from the long-term trend (Jenkins et al., 2018). The origin of the persistent long-term forcing is unknown; one plausible explanation is that changes in Antarctic zonal winds have

enhanced the influxes of warmer waters underneath the ice shelves (Buizert et al., 2018), a phenomenon that has been linked to anthropogenic warming and teleconnections with tropical Pacific variability (Dutrieux et al., 2014; Sadai et al., 2020; Paolo et al., 2018; Holland et al., 2019).

To analyze the temporal evolution of changes, we computed area-averaged time series of thickness for each ice shelf (excluding regions within 6 km of calving fronts or within 3 km of grounding lines) and neighboring locations

upstream of the grounding lines (Figure 13). To highlight the patterns over the deepest portion of the ice shelves and avoid any influence of advancing calving fronts, we show time series for the thickest 50% ice of each ice shelf. All ice shelves in the Amundsen and Bellingshausen Sea sectors show dramatic rates of thinning since records began in the early 1990s, with thickness losses as high as $6.1 \pm 0.5$ m yr$^{-1}$ (Crosson) over the full ice shelf, confirming previous findings (Jenkins et al., 2018, 2010; Smith et al., 2017; Shepherd et al., 2004; Pritchard et al., 2012) that the dominant

driver of change predates the satellite era. Over the 26-year record some ice shelves have thinned by more than 20% near their original grounding lines, e.g. Thwaites: 23% (right before calving circa 2013), Crosson: 35% and Dotson: 25% (Figure 13).

Our data reveal a large-scale coherent pattern of slowdown in rates of ice shelf thinning (Figure 11) towards the end of the record, starting around 2008. The slowdown is most pronounced for those ice shelves that have thinned most

over the 26-year period (Figure 15), and is particularly accentuated along the West Antarctic margin where ocean melting of ice shelves is strongest (Figure 12), but also occurs along Wilkes Land in East Antarctica. For the Amundsen Sea ice shelves, slowdown in thinning began between 2006 (Pine Island) and 2009 (Dotson), and at a later epoch in the Bellingshausen Sea sector (~2010, Venable), somewhat consistent with previous single-ice-shelf studies (Jenkins et al., 2018; Davis et al., 2018). In many cases, the slowdown signal is sufficient to offset previous

acceleration; with average deceleration reaching up to $-22 \pm 3$ cm yr$^{-2}$ (or $-51 \pm 7$ cm yr$^{-2}$ near the GL) for Crosson Ice Shelf. Overall, we estimated an average thinning slowdown of $8.3 \pm 1.3$ and $1.1 \pm 1.0$ cm yr$^{-2}$ for the ASE and BSE, respectively.

Thinning of grounded ice (Nilsson et al., n.d.) between 3 and 12 km upstream of the grounding line (Figure 13) shows that the grounded ice losses have increased on average in West Antarctica as a dynamical response of glacier flow to

loss of ice-shelf buttressing and grounding line retreat (Konrad et al., 2017; Gudmundsson et al., 2019; Milillo et al., 2022). More recently, however, the rate of grounded ice loss appears to be responding to the slowdown in the rate of ice shelf thinning, as suggested by the coincident change in the trend of both floating and grounded thickness change records (Figure 13). Grounded ice flow is also controlled by factors other than ice shelf thickness (i.e. changes in ice shelf front, grounding line position, pinning points, and basal friction), and changes in ocean hydrographic properties



may ultimately dictate how the ice sheet will change over the coming centuries. The longer-term response of glacier flow to the slowdown in ice-shelf thinning in the region remains to be confirmed.

The slowdown in thinning rates implies a recent decrease in the rate of ice shelf mass loss. Satellite measurements of ice thickness alone cannot directly indicate whether the slowdown in thinning rates reflect an increase in grounded ice flux, an increase in surface accumulation, or a decrease in basal melt rates. To investigate this, we constructed time-
variable velocity fields for the Amundsen Sea sector by combining satellite radar and optical velocity products (Mouginot et al., 2014, 2017a; Gardner et al., 2018). We found that substantial changes in ice flow occurred on both the floating and grounded ice (Mouginot et al., 2014; Konrad et al., 2017), with velocity increases over the ice shelves up to four times that of the grounded glaciers that feed them. A widespread ice flow acceleration across the grounding line in the Amundsen Sea sector after the 2000s (Mouginot et al., 2014) provided a sudden influx of ice, decreasing
ice shelf thinning rates by over 1.5 m yr$^{-1}$, which accounted for about 75% of the slowdown in the observed rate of ice shelf thinning in Pine Island and Thwaites. The dynamic contribution to thinning deceleration was less, but still substantial, in the case of Crosson (1.1 m yr$^{-1}$ or 21%) and significantly less for Dotson (~0.2 m yr$^{-1}$ or 7%) (Table 2). Changes in ice flow alone are, therefore, unable to fully explain the observed slowdown in thinning rates. An investigation of changes in surface mass balance shows relatively minor trends in surface accumulation over the
observation period (Figure 2), with little impact on the overall rates of ice shelf thickness change (accounting for 1–2%, Table 2), leaving changes in ocean melt rates as the most likely contributor to the recent slowdown in thinning rates.

To investigate the role of changes in ocean induced melting, we calculated time-variable basal melt, after accounting for time-varying ice flow and surface effects. Our results show that rates of ice shelf basal melt have systematically
decreased near the grounding lines, by varying degrees, from the late 2000s to 2017. This reduction in basal melt is greatest in the regions of West Antarctica that have been changing most rapidly, such as Pine Island, Thwaites, Crosson, Dotson and Venable grounding lines (Figure 16). These ice shelves are also the thickest, with their deep grounding lines exposed to intrusions of mCDW through bathymetric throughs, in contrast to shallower ice shelves in the region that do not exhibit a clear reduction in melt rate (e.g. Abbot and Cosgrove). The largest decreases in basal
melt rate, over the 26-year period, are found near the grounding lines of Pine Island and Crosson ice shelves, with -59 ± 4. and -84 ± 6 cm yr$^{-2}$, respectively. On some ice shelves, such as Venable and Stange, slowdown in thinning and basal melting is strictly confined to the floating ice near the grounding line (Figure 12). For Venable, this results in a slight overall increase in thinning/melting, on average, for the full ice shelf extent (Figure 11 and Table 3), but substantial decrease near the grounding line: -7 ± 2 cm yr$^{-2}$ reduction in meltwater rate (Figure 16). One factor limiting
estimates of time-dependent basal melt rates is the lack of time-variable velocity information. In general, ice shelves outside the Amundsen sector have not experienced dramatic velocity changes (Rignot et al., 2019; Gardner et al., 2018). Assuming a constant velocity field, however, can bias (high) estimates of basal melt change of ice shelves such as Getz, known to have had significant changes in velocity (Gardner et al., 2018; Selley et al., 2021). Still, modest velocity changes such as those observed on Dotson Ice Shelf (Figure 7) only contributed about 7% to the total change
in thinning rates (Table 2).



Ice shelf basal melt rates are sensitive to changes in the ice shelf draft depth, which dictates the temperature of the ocean waters that come into contact with the ice shelf base (Padman et al., 2012; Schodlok et al., 2012). To test the ability of our thickness product to resolve basal melt structures, we evaluated the effects of changes in sub-ice-shelf cavity geometry by modeling basal melt rates for prescribed ice shelf thickness and ice front positions for the years

1993 and 2017. In this (simplistic) sensitivity experiment, we applied the surface forcing and ocean conditions corresponding to either year on each setup (Table 1). The model output shows a highly-resolved basal melt field (Figure 17). When forcing is held constant, melt rates for the 2017 geometry are generally lower than for the 1993 geometry near all grounding lines, with some localized patches of higher melt elsewhere on the ice shelves (Figure 17). Modeled basal melt rates are 25% to 50% lower near the grounding lines of the Dotson and Crosson ice shelves

when using the 2017 ice shelf geometry compared to rates determined form the 1993 geometry. This melt reduction at depth is consistent with the notion that inflows of mCDW into the sub-ice-shelf cavities may be counteracted by a thinned ice shelf whose draft sits in shallower (cooler) waters; or with the idea that cold meltwater from the deep grounding lines might reduce melt at shallower depths. Other investigations, however, have suggested that in the early 2010s ocean conditions in the ASE changed, further contributing to a reduction in basal melt of the West Antarctic

ice shelves (Jenkins et al., 2018; Webber et al., 2017). We note that our modeling experiments do not negate the influence of changes in thermocline depth on ice shelf melt, or is intended to single out a driving mechanism (due to its simplicity), but rather suggest that changes in cavity-geometry play a significant role in melt variability.

The link between ice shelf thinning and ice shelf melt comprises a negative feedback relationship that acts in tandem with a separate negative feedback relationship between grounded ice acceleration and ice shelf thinning: (i) Prolonged

ice shelf thinning and grounding line retreat have reduced the backstress that ice shelves provide, allowing outlet glaciers to accelerate (Konrad et al., 2017; Gudmundsson et al., 2019; Minchew et al., 2018), and the new influx of grounded ice has provided some mitigation to the overall thinning rate of ice shelves in the ASE and BSE. (ii) Thinning may have also led to a reduction in basal melt rates by placing ice shelf drafts in cooler waters compared to their geometric configuration in the 1990s and early 2000s (Padman et al., 2012). We hypothesize that these two feedback

mechanisms account for the majority of the recent slowdown in ice shelf thinning, with the remainder attributable to a multi-year reduction in basal melt due to a change in hydrographic properties (e.g. a temporary shift in the depth of the thermocline).

## 4 Conclusions

We have examined in detail the time-varying evolution of Antarctic ice shelf thickness over a 26-year period. We

show overall thinning around Antarctica consistent with previous studies, but also that there has been a significant and consistent slowdown in thinning since around 2008 across several West Antarctic and Wilkes basin ice shelves. We suggest that the slowdown in melt that we observe might result from complex feedback mechanisms that can modulate rates of ice shelf thinning. We note that our observations span only 26 years, and the reduction in thinning we report may represent a temporary adjustment period on decadal timescales. We neglect areas within 3 km of the grounding

line to limit the influence of bridging stresses; yet, our measurements could still be influenced by local transient



changes in the hydrostatic state of ice within our areas of observation. The melt rates we report could also be influenced by changes in grounding line position that occur outside our region of analysis, as hotspots of melt migrate nearer or farther from our fixed mask. Nonetheless, the slowdown in melt that we report is seen across several ice shelves, and in multiple sectors of Antarctica.


The apparent feedback mechanisms we observe are not currently included in the pan-Antarctic ice sheet models that have been used to generate sea level projections for the coming century (Seroussi et al., 2020; Fox-Kemper et al., 2021). We cannot offer a complete picture of the long-term impact of the processes we describe until they are adopted in fully coupled pan-Antarctic ice-ocean models. However, our findings indicate that by including ice dynamic

feedbacks and the tendency for ice shelves to thin themselves into cooler waters, projections of ice loss may prove more complex, and possibly more tempered than current estimates suggest.

**Data and Code availability**

All ice shelf thickness and basal melt rate data generated in this study are freely available through the NASA MEaSUREs ITS_LIVE project (https://its-live.jpl.nasa.gov). All the code developed to process and analyze the satellite data used in this study is freely available as an open-source Python package hosted on GitHub (https://doi.org/10.5281/zenodo.3665785). The Glacier Energy and Mass Balance model (GEMB) used for firn and surface mass balance modeling is a module of NASA's open-source Ice-sheet and Sea-level System Model (ISSM).

All model outputs and the models themselves are freely available upon request.

**Acknowledgements**

The authors were supported by the ITS_LIVE project awarded through NASA MEaSUREs program, and the NASA Cryosphere program. We thank the European Space Agency (ESA) for distributing their radar altimetry data. We

thank Dr. T. Armitage of JPL for providing the Antarctic mean sea level data. We thank Dr. A. Thompson of Caltech for useful discussions on ice-ocean interactions. We thank the JPL Supercomputing Group for assisting with HPC resources. The research described in this paper was carried out at the Jet Propulsion Laboratory, California Institute of Technology, under a contract with NASA.

**Author contributions**

F.S.P and A.S.G. conceptualized the study. F.S.P. processed the ice shelf data, developed the melt rate inversion and performed the analyses. C.A.G. performed the quality assessment. J.N.N. processed the grounded ice data. F.S.P and J.N.N. developed the method and code for altimetry data. M.P.S. setup and ran the ice-ocean model. A.S.G. and N.S. developed the firn model and N.S. ran the model. F.S.P, A.S.G., C.A.G. and H.A.F. wrote the majority of the main

text. All authors contributed to the writing and editing of the manuscript.



**Competing interests**

The authors report no competing interests.

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





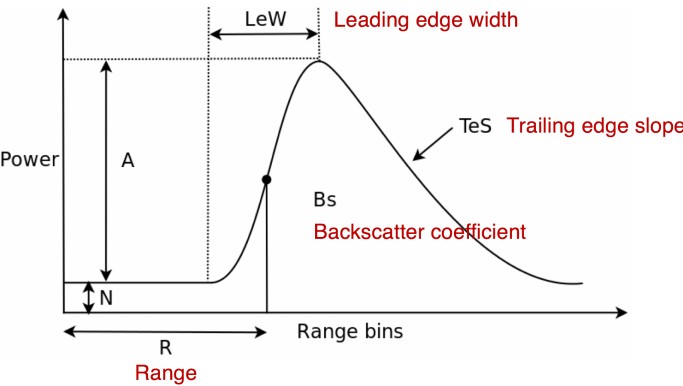

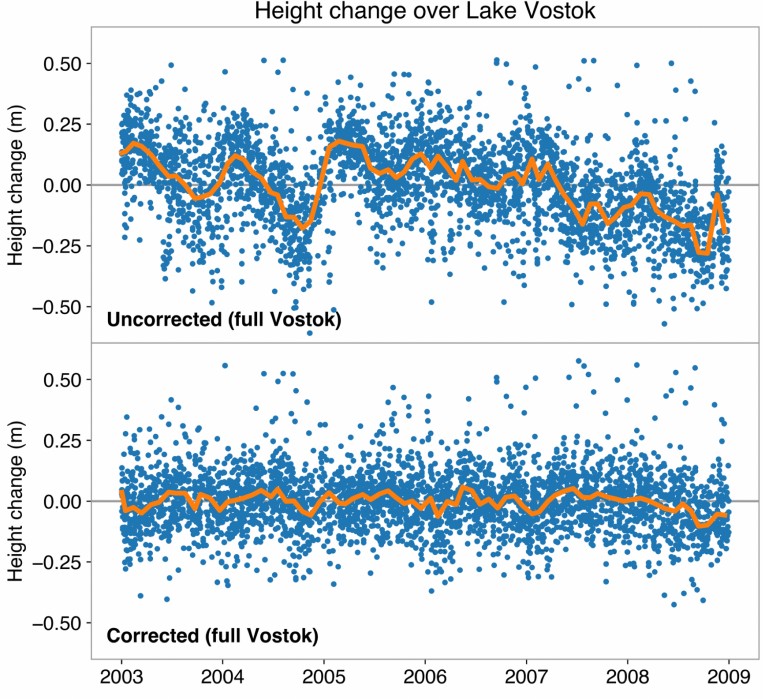

**Figure 1: Multi-parameter radar scattering correction.** (top) The different waveform parameters used to characterize the radar echo (where A is amplitude, N is the noise floor, and 'Range bins' are the discrete samples of the return signal). (bottom) Time series of individual point high measurements before and after applying the scattering correction. The example shows Lake Vostok where GPS records show no significant trend or variability in surface height.





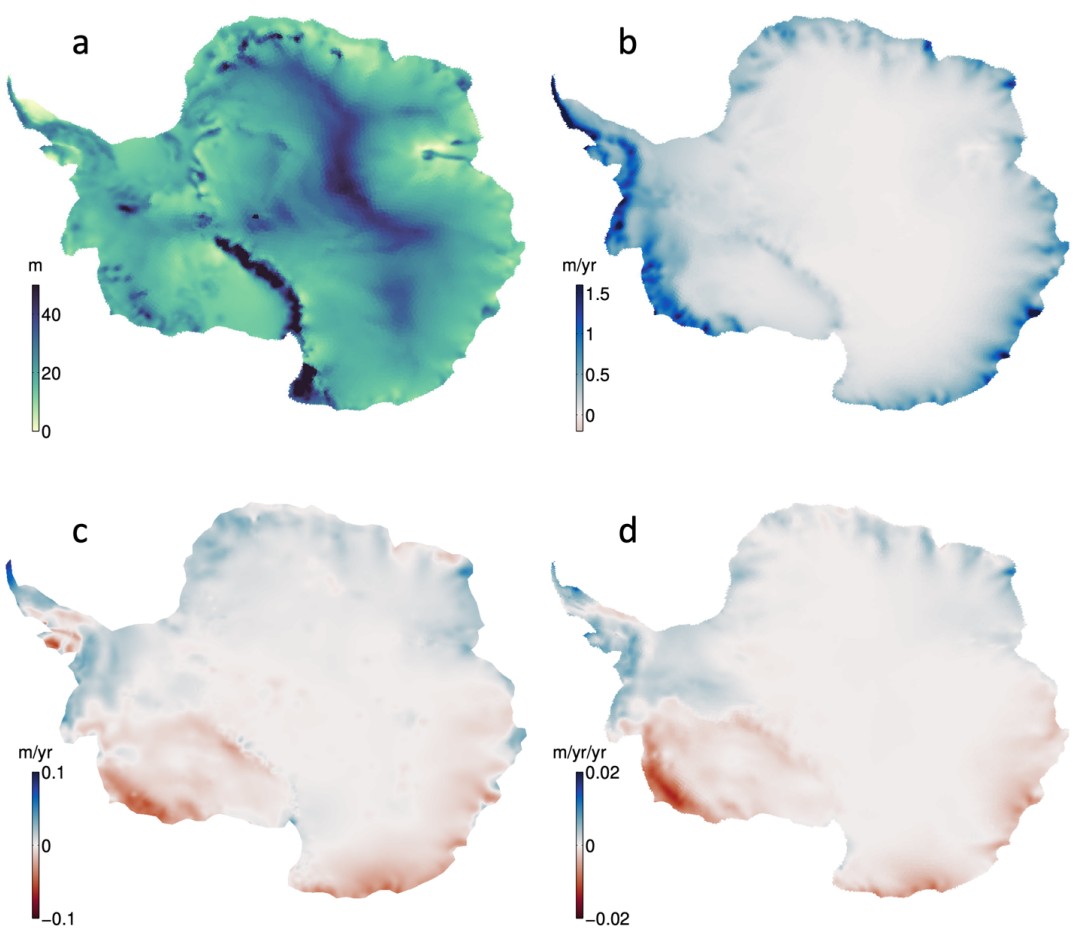

**Figure 2: Glacier Energy and Mass Balance Model (GEMB).** The 1992—2017 mean (a) firn air content (FAC) and (b) surface mass balance (SMB) simulated by GEMB forced with ERA5 reanalysis data, accompanied by the corresponding simulated trends in (c) FAC and (d) SMB over the same period.







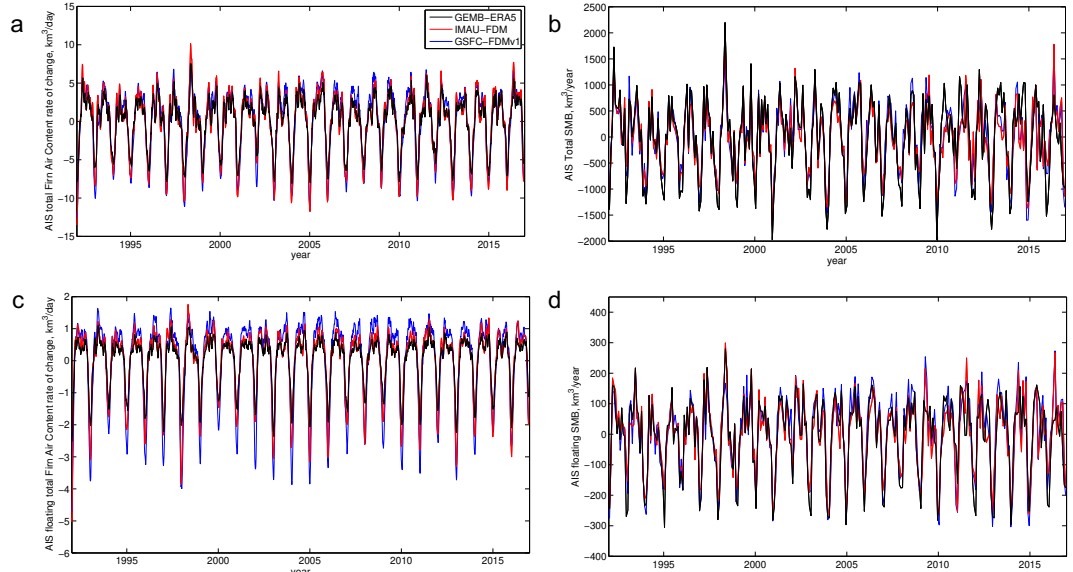

**Figure 3: Glacier Energy and Mass Balance Model (GEMB) timeseries comparison.** The 1992-2017 monthly running mean of (a,c) rate of FAC volume change and (b,d) SMB simulated by GEMB (black), compared with estimates from IMAU-FDM RACMO2.3 (red) and GSFC-FDMv1 (blue) summed over the entire ice sheet (a,b) and over floating ice (c,d).

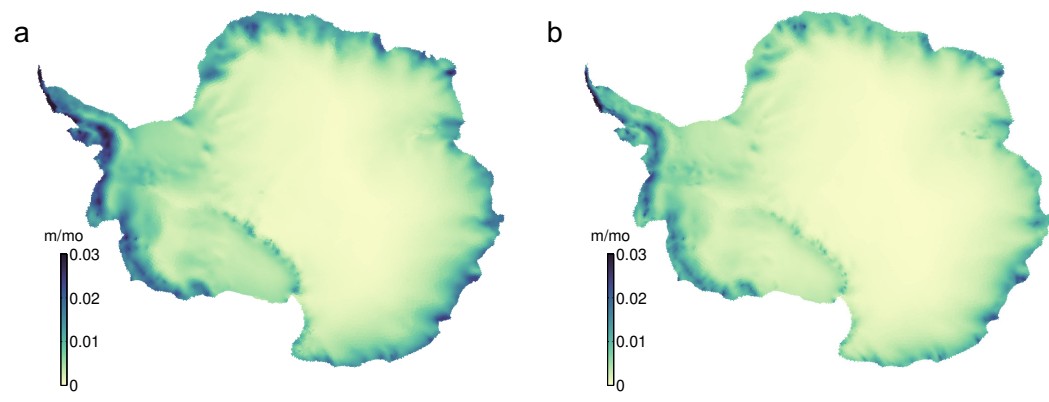

**Figure 4: Firn air content and surface mass balance error estimates.** The 1992-2017 mean of the 5-month timeseries for (a) error in rate of change in FAC and (b) error in SMB.



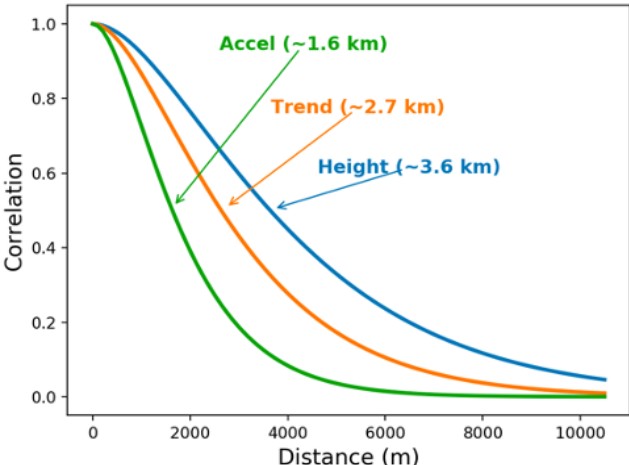


**Figure 5: Decorrelation lengths for height, trend and acceleration.** Example of spatial correlation functions (or covariance normalized) used to derive continuous fields for each ice shelf parameter, without imposing a (single) spatial resolution *a priori* to all components. Shown in this example are typical spatial scales for each quantity derived 925   from satellite radar altimetry over the Ross Ice Shelf.






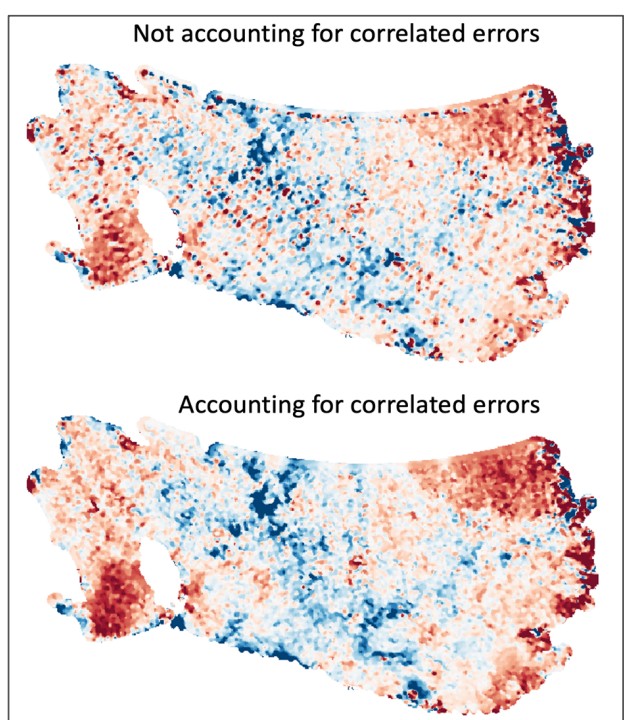

**Figure 6: Optimal Interpolation with correlated errors.** Example of our spatial Optimal Interpolation approach (one timestep) where we account for the satellite along-track correlated errors. (top) Points along the same ground tracks share the same long-wavelength errors, producing a "track pattern" in the interpolated fields (which is the case in most standard interpolation approaches). (bottom) When correlated errors are accounted for (as off-diagonal elements in the error matrix) points sharing the same correlated errors are downweighed, producing a coherent spatial field.






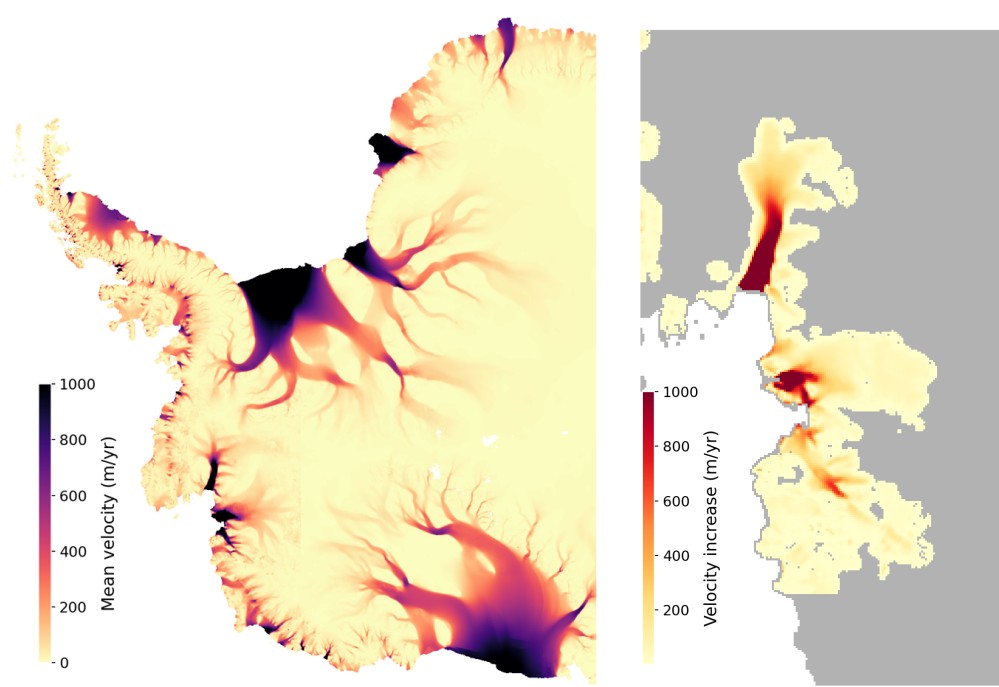


**Figure 7: Mean ice shelf velocity and changes in Amundsen ice flow.** Reference ice velocity (left) and velocity change from 1996 to 2017 (right) used in this study. Ice shelf velocity changes in order of magnitude: Pine Island, Thwaites calved, Crosson, Thwaites remnant, and Dotson with only a minor change.






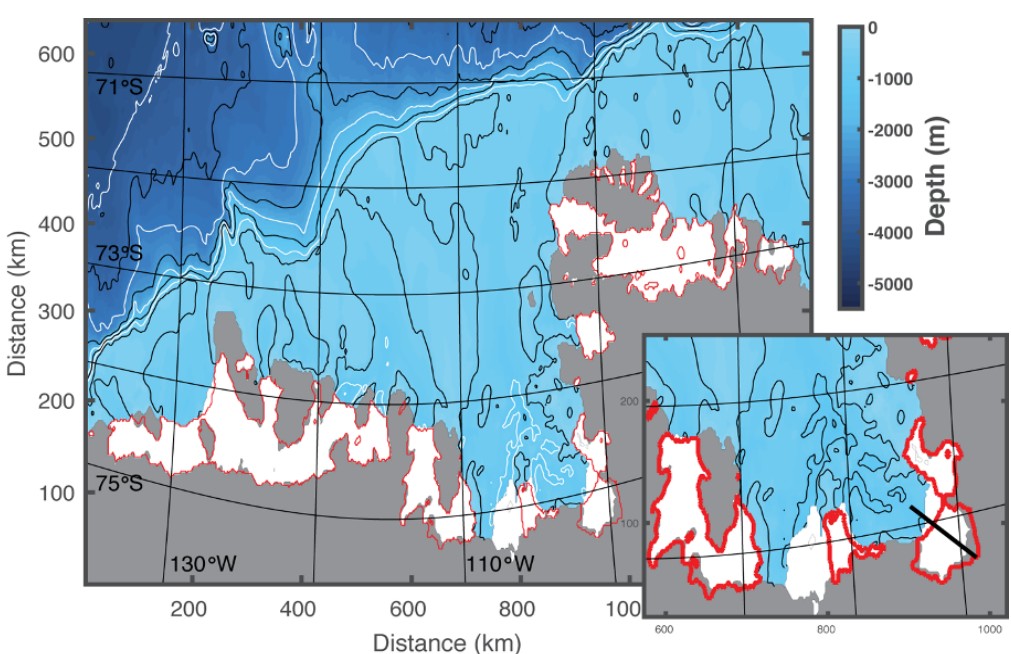


**Figure 8: Amundsen Sea model domain at 3-km grid spacing**. The Antarctic Continent is depicted in grey, ice shelf area for 1993 in white, and red contours the ice shelf extent in 2017. Contour lines of bathymetry from 500 m in 500-m intervals are shown in black, and from 1000 m in 1000-m intervals in white. Note the slight distortion of the LLC grid from a true Latitude/Longitude grid, ensuring little deviation from an isotropic grid. The inset shows Pine Island, Thwaites, Crosson and Dorson ice shelves where grounding line and ice shelf front retreat is prominent.







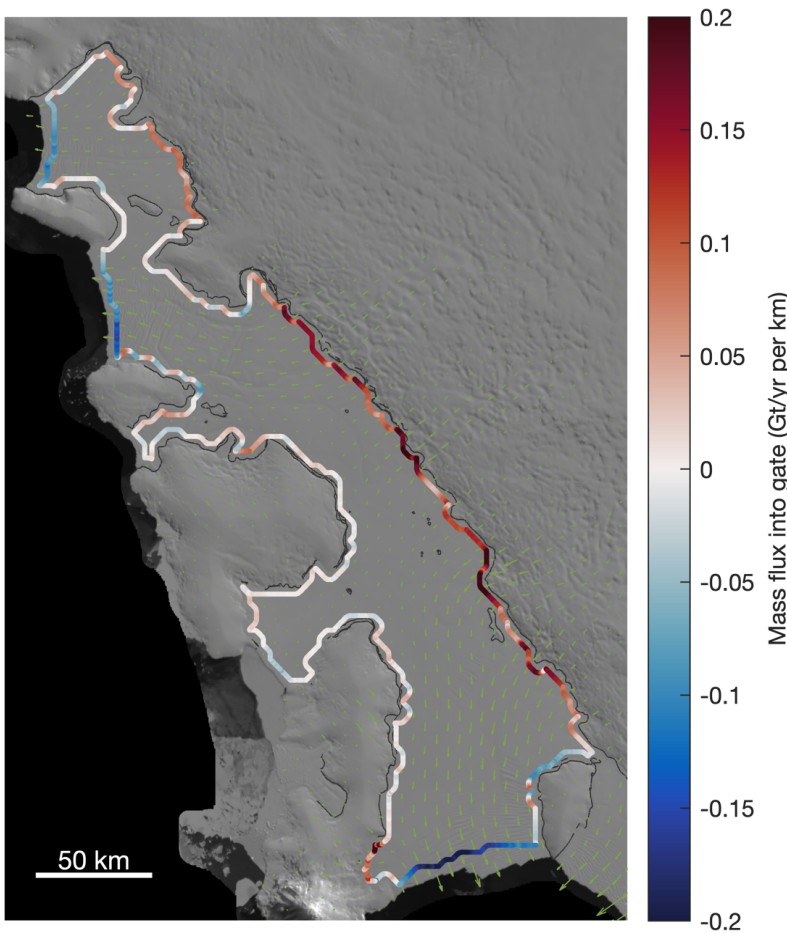

**Figure 9: Control volume delineation for Getz Ice Shelf.** Example of ice flowing into (red) and out (blue) of a control volume. For each ice shelf, we define a control volume as the largest hydrostatically floating area within previously published ice shelf outlines (*Mouginot et al., 2017*), which we buffer inward by 3 km. In this figure, green vectors show ice velocity from ITS_LIVE data and thin black lines show grounding lines obtained by InSAR (*Rignot et al., 2016*).





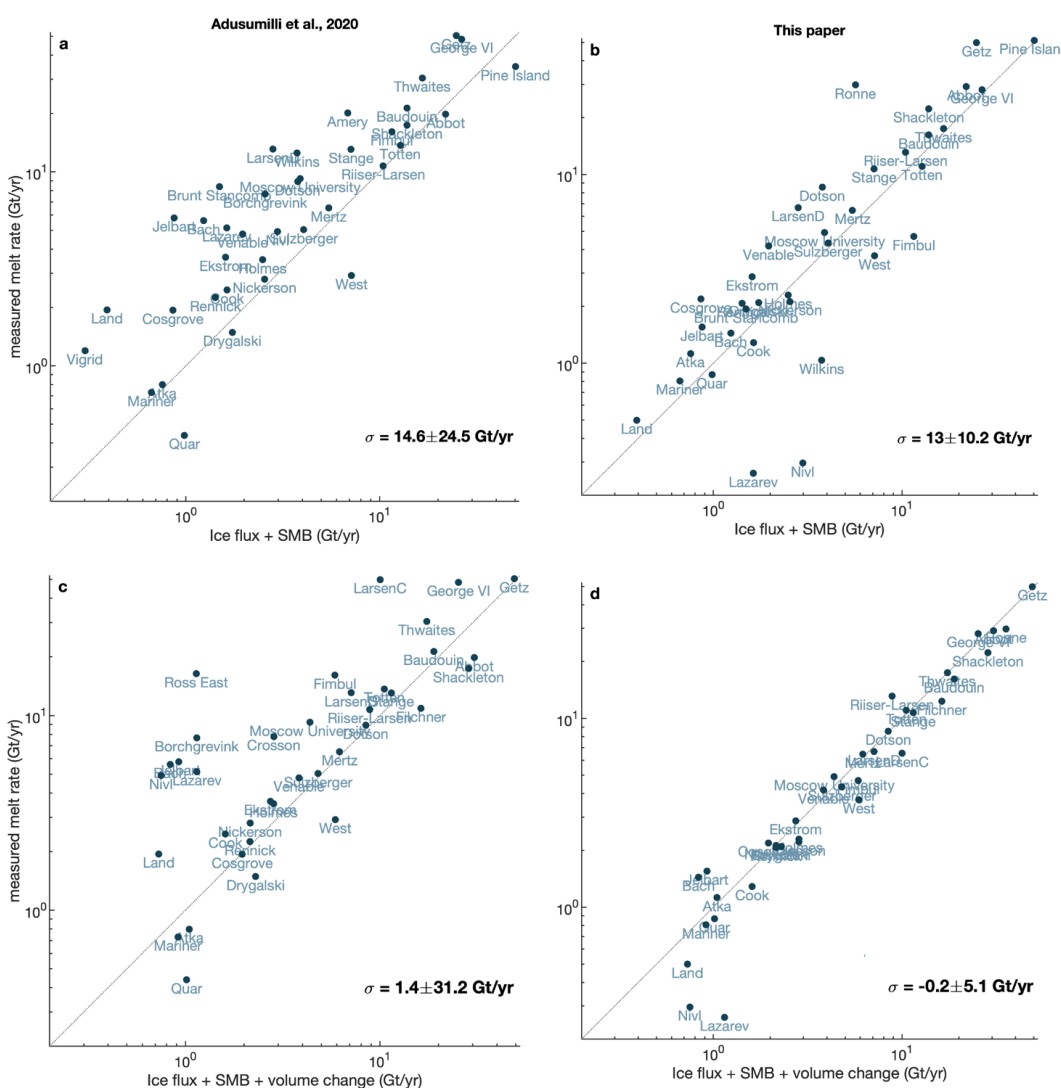


**Figure 10: Comparison of estimates from this study with previous work.** Comparison of (left) *Adusumilli et al. (2020)* and (right) this study's ice shelf basal melt estimates against a control-volume calculation of ice shelf mass change (line). The control volume is based on the input and output fluxes across the grounding line and ice front, mass gained or lost due to surface mass balance (top row), and ice loss due to anomalies in basal melt (bottom row). For this comparison, we only considered grid cells that are at least 90% hydrostatically compensated (near fully floating).



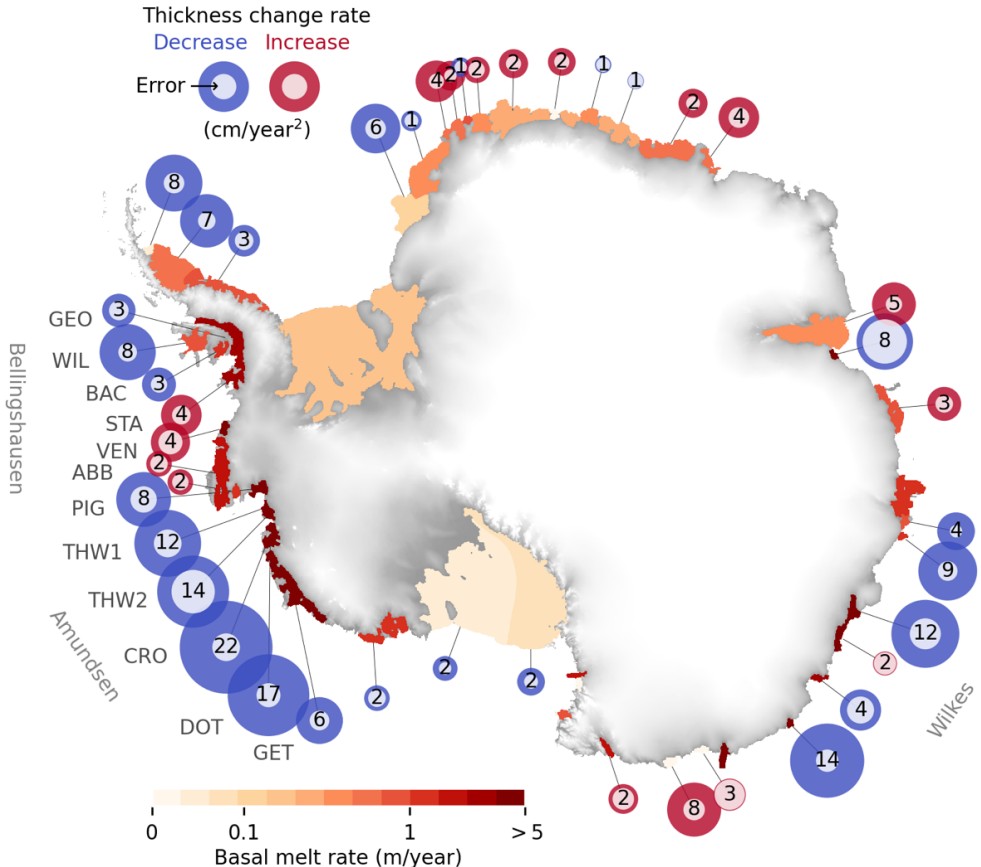

**Figure 11: Ice shelf thinning has slowed where basal melt rates are highest.** Circum-Antarctic pattern of acceleration/deceleration in ice shelf thickness change rate (circles) and basal melt rate (field) for each ice shelf. Blue circles represent a decrease, on average, in the rate of thickness change. Inner circles represent the respective uncertainties. All values are the 26-year means (1992—2017) averaged over the respective ice shelf areas (values near the grounding lines for the ASE in Sup. Fig. 1). Values are rounded for visualization purposes (see mean values with respective uncertainties and original precision in Table 3; and spatial field at full resolution with corresponding error distribution in Sup. Fig. 5). Basal melt rates are displayed in logarithmic scale. Non-significant values are omitted from the plot.



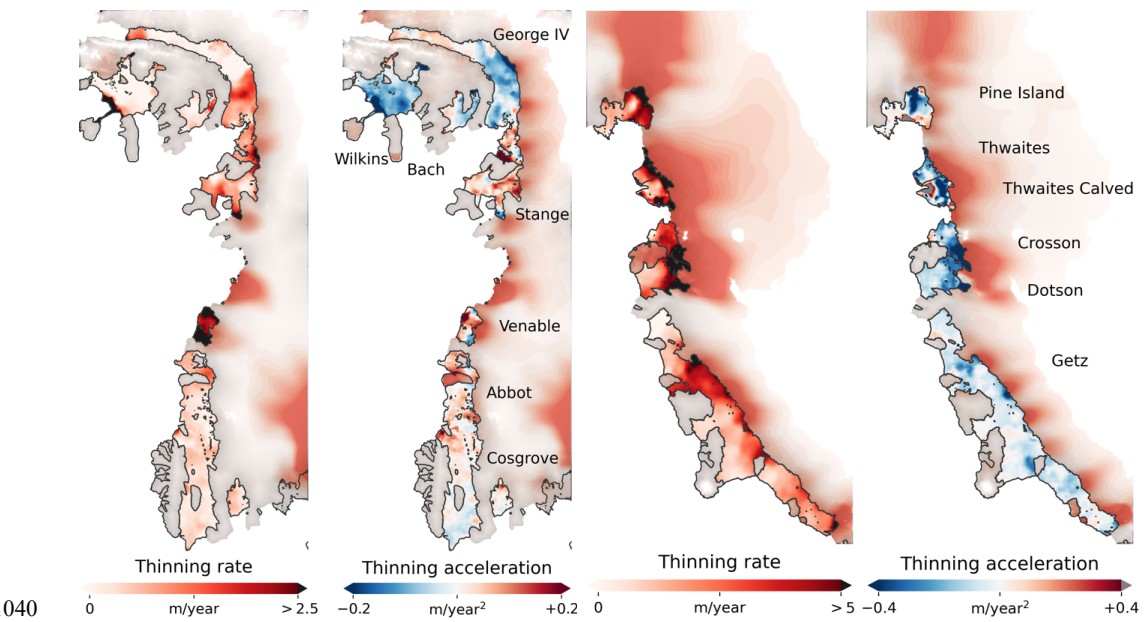


**Figure 12: Slowdown in thinning is more pronounced where ice shelves are thicker.** Panels (a) and (c): Ice shelf and grounded ice thinning rate (in meters of ice equivalent per year); Panels (b) and (d): Mean acceleration (positive values, red) and deceleration (negative values, blue) in ice shelf and grounded ice thinning. Values are the mean over 1045 the 26-year period. The color bars depict ice shelf values; for reference, grounded ice values are an order of magnitude smaller than ice shelf values. Calved areas shown (e.g. Wilkins, Pine Island and Thwaites ice shelf fronts) were excluded in all calculations.






**Figure 13: Thinning rates have slowed in most recent decade across ice shelves.** [top] (left) Cumulative ice shelf thickness change for ice shelves with the highest losses in West Antarctica. (right) Respective rate of thickness change, with mean rate values for highlighted time intervals on each side (white/gray area). Ice shelf time series are averages over the area above the mean thickness value (i.e. the 50% thickest ice). Unsmoothed time series with error bars and statistical significance of linear trends are shown in Sup. Fig. 1. [bottom]. The same as the top panels but a few kilometers upstream of the grounding lines of the respective ice shelves.




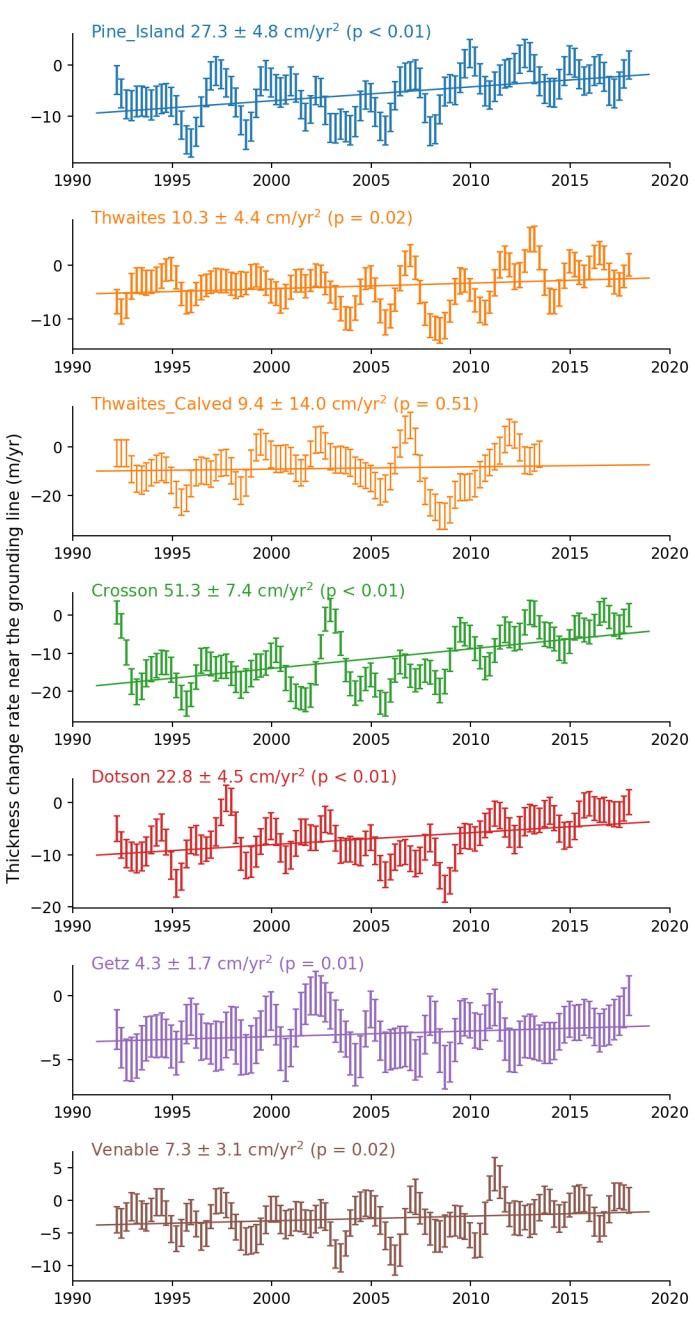

**Figure 14: Time series of instantaneous rate of change in ice shelf thickness**, unsmoothed with respective error bars, least-squares trend and the associated statistical significance. A positive trend means deceleration in the thinning rate.







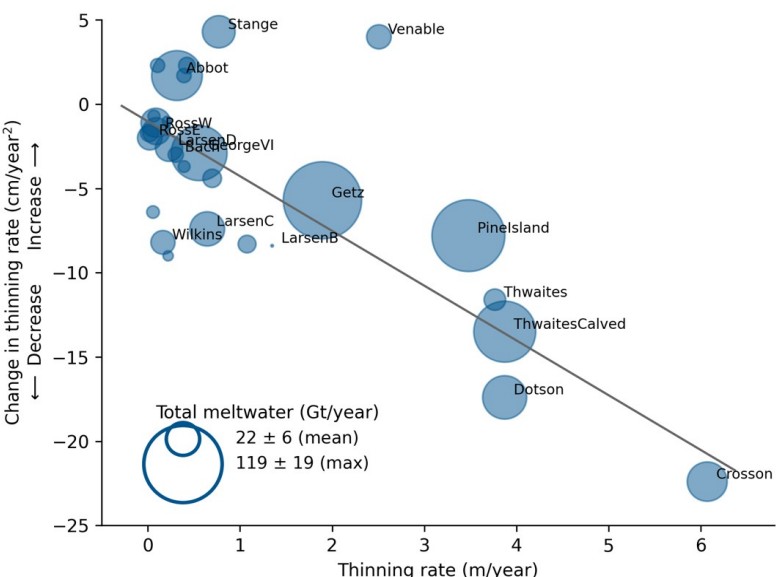

**Figure 15: Thickness loss and slowdown in thinning are strongly correlated.** Relationship between rates of ice
shelf thinning and change in thinning rates around Antarctica. Only ice shelves with statistically significant
acceleration (positive change) or deceleration (negative change) in thinning are displayed (30 ice shelves). Circle areas
are proportional to the total meltwater produced (basal melt rate multiplied by ice shelf area), with Getz showing the
largest value. The regression line has a *correlation coefficient* of $\rho$ = -0.74 ($\rho$ = -0.84 without Stange and Venable)
with a *p-value* < 0.01. (All quantities with respective uncertainties are presented in Table 3)









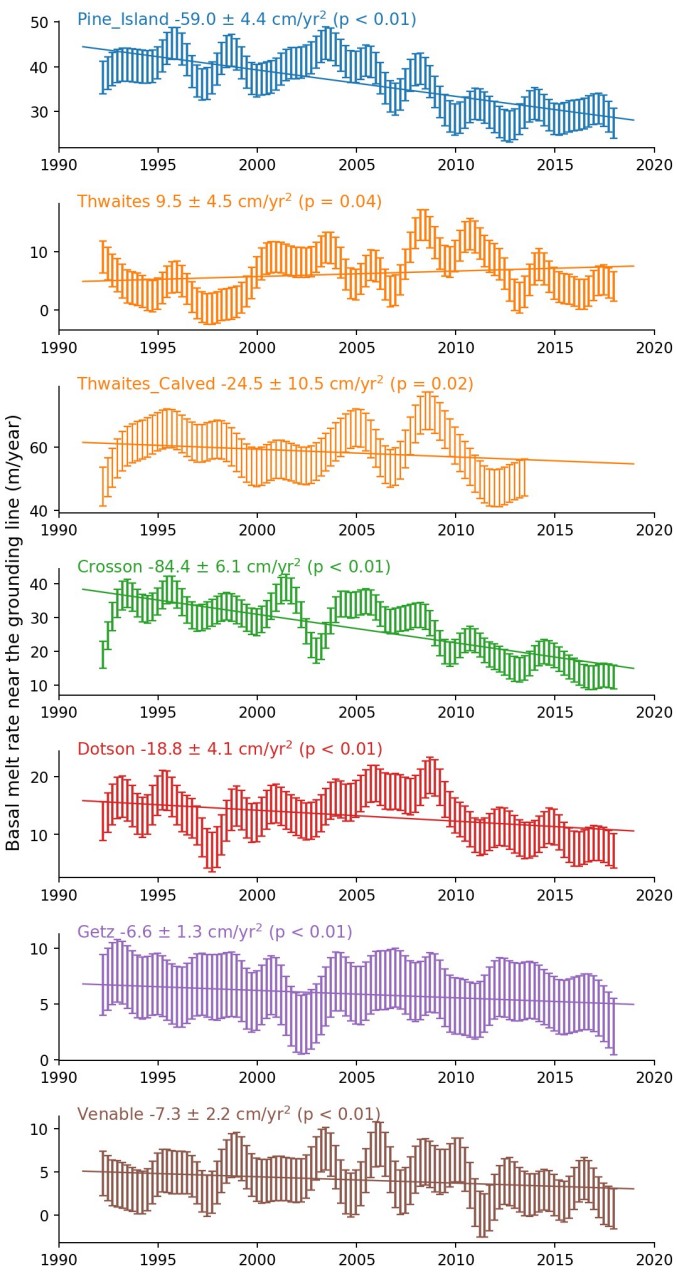

**Figure 16: Basal melt rates have abated around the Amundsen Sea.** Time series with error bars of basal melt rate (in meters of ice equivalent per year) for the thickest portions of the ice shelves, i.e. the area with thickness above the mean thickness value. Trend line (acceleration/deceleration) is a least-squares fit with respective statistical significance.





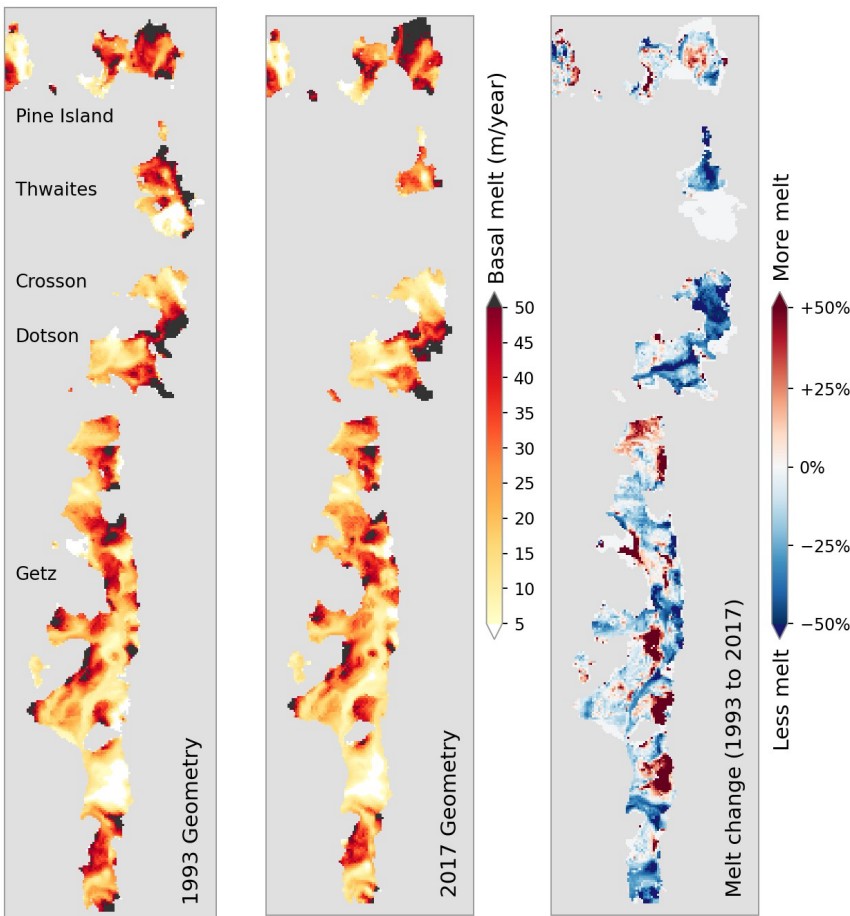

**Figure 17: Satellite ice shelf thickness yields high-resolution modeled basal melt rates.** Holding ocean temperatures constant at the "warm" conditions observed in 2017, modeled melt rates for ice shelf draft and front positions corresponding to 1993 (A) and 2017 (B), show an overall reduction in melt rates (C) resulting from changes in ice shelf geometry alone (See Table 1). Results using "cold" ocean conditions of 1993 show a similar pattern, suggesting that the effects of changing ice shelf geometry occur regardless of changes in ocean temperature. Insignificant melt rate values (< 0.1 m/year) and non-overlapping areas between 1993 and 2017 are masked out (white).



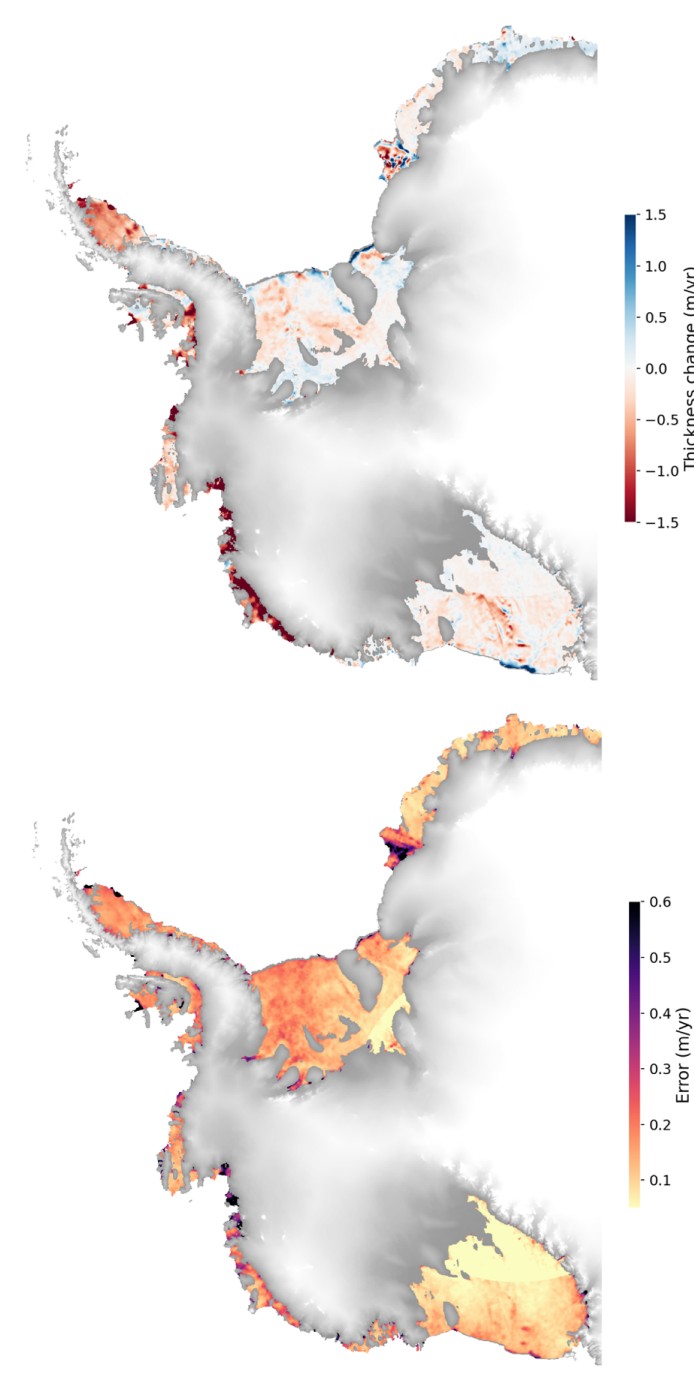


**Figure 18: Satellite derived thickness change field and associated error.** Ice shelf thickness change rate (top) and associated error (bottom) from which basal melt rates are estimated.





|  | Hydrography | |
|---|---|---|
| **Draft** | Cold (1993) | Warm (2017) |
| Deep (1993) | DC 93/93 | DW 93/17 |
| Shallow (2017) | SC 17/93 | SW 17/17 |

**Table 1: Model Simulation Abbreviations.** The four model simulations are determined by the depth of the ice shelf draft (deep-1993 and shallow-2017) and the hydrographic properties on the continental shelf and in the ice shelf cavities (cold-1993 and warm-2017).


| Name | Delta Thin | Delta Div | Delta SMB | (Div %) | (SMB %) |
|---|---|---|---|---|---|
| Thwaites | 2.24 | 1.75 | 0.03 | 78.00 | 1.00 |
| Pine_Island | 2.00 | 1.50 | 0.02 | 75.00 | 1.00 |
| Dotson | 2.23 | 0.15 | 0.05 | 7.00 | 2.00 |
| Crosson | 4.93 | 1.05 | 0.03 | 21.00 | 1.00 |


**Table 2: Contributions to change in thinning rate for major Amundsen Sea ice shelves (1992—2008 to 2009—2017).** Percentage contributions to the observed change in ice shelf thinning rate from ice flux divergence, which includes ice advection and stretching (dynamic thinning), and surface mass balance. Units are meters of ice equivalent per year (m/yr). Only ice shelves with consistent availability of time-evolving velocity (needed to estimate time-1135 evolving divergence) are shown. These quantities are the change in the mean values from 1992—2008 to 2009—2017, averaged over the respective ice shelf areas. Note that if the averaging area is restricted to the thickest ice (i.e. areas close to the grounding lines), the contribution from basal melting is significantly larger (see Figures 2 and 3).






| Ice shelf | Area (km$^2$) | Thinning rate (m/year) | Basal melt rate (m/year) | Thinning accel. (cm/year$^2$) |
|---|---|---|---|---|
| Ronne | 327,407.1 | 0.07 ± 0.15 | 0.14 ± 0.20 | -0.8 ± 0.8 |
| LarsenE | 1,167.3 | 0.14 ± 0.49 | 0.94 ± 0.51 | 0.4 ± 2.7 |
| LarsenD | 22,594.3 | 0.23 ± 0.18 | 0.69 ± 0.20 | -2.6 ± 1.0 |
| LarsenC | 47,443.5 | 0.65 ± 0.17 | 0.54 ± 0.20 | -7.4 ± 1.0 |
| LarsenB | 2,151.1 | 1.35 ± 0.28 | 0.06 ± 0.43 | -8.4 ± 1.5 |
| George VI | 23,259.9 | 0.56 ± 0.18 | 2.85 ± 0.24 | -2.9 ± 1.0 |
| Wilkins | 12,906.7 | 0.17 ± 0.23 | 0.95 ± 0.28 | -8.2 ± 1.3 |
| Bach | 4,547.9 | 0.30 ± 0.26 | 1.08 ± 0.29 | -3.0 ± 1.4 |
| Stange | 7,930.0 | 0.77 ± 0.19 | 2.86 ± 0.27 | 4.3 ± 1.1 |
| Venable | 3,155.0 | 2.51 ± 0.31 | 4.04 ± 0.37 | 4.0 ± 1.7 |
| Abbot | 29,351.7 | 0.32 ± 0.16 | 1.85 ± 0.19 | 1.7 ± 0.9 |
| Cosgrove | 2,989.5 | 0.39 ± 0.17 | 1.45 ± 0.21 | 1.7 ± 0.9 |
| Pine Island | 6,120.3 | 3.48 ± 0.46 | 18.35 ± 3.40 | -7.8 ± 2.5 |
| Thwaites | 1,772.0 | 3.77 ± 0.46 | 5.57 ± 4.01 | -11.6 ± 2.5 |
| Thwaites Calved | 3,116.7 | 3.87 ± 1.09 | 26.01 ± 8.52 | -13.5 ± 6.0 |
| Crosson | 3,331.2 | 6.07 ± 0.54 | 10.11 ± 2.04 | -22.4 ± 3.0 |
| Dotson | 5,677.3 | 3.87 ± 0.34 | 7.23 ± 0.96 | -17.4 ± 1.9 |
| Getz | 32,783.5 | 1.90 ± 0.20 | 3.95 ± 0.34 | -5.7 ± 1.1 |
| Nickerson | 6,335.3 | 0.02 ± 0.18 | 1.07 ± 0.24 | -1.7 ± 1.0 |
| Sulzberger | 11,969.5 | 0.05 ± 0.18 | 1.33 ± 0.21 | -0.1 ± 1.0 |
| Ross West | 293,551.9 | 0.09 ± 0.08 | 0.05 ± 0.10 | -1.6 ± 0.4 |
| Ross East | 186,876.5 | 0.02 ± 0.10 | 0.07 ± 0.14 | -2.0 ± 0.5 |
| Drygalski | 2,280.9 | 0.13 ± 0.47 | 2.05 ± 0.64 | -1.8 ± 2.6 |
| Nansen | 1,942.1 | -0.15 ± 0.32 | -0.14 ± 0.34 | 0.1 ± 1.8 |
| Mariner | 2,672.9 | -0.04 ± 0.20 | 0.87 ± 0.22 | -0.7 ± 1.1 |
| Rennick | 3,276.5 | 0.42 ± 0.25 | 1.76 ± 0.27 | 2.3 ± 1.4 |
| Cook | 3,531.8 | -0.58 ± 0.24 | -0.15 ± 0.33 | 7.6 ± 1.3 |
| Ninnis | 1,929.0 | 0.07 ± 0.47 | -5.79 ± 0.69 | 2.7 ± 2.6 |
| Mertz | 5,652.2 | -0.03 ± 0.23 | 4.42 ± 0.32 | -0.9 ± 1.3 |
| Dibble | 1,463.2 | -0.32 ± 0.31 | 6.42 ± 0.35 | -14.3 ± 1.7 |
| Holmes | 2,366.4 | 0.70 ± 0.38 | 3.07 ± 0.46 | -4.4 ± 2.1 |
| Moscow University | 5,949.6 | 0.31 ± 0.27 | 4.57 ± 0.35 | 1.5 ± 1.5 |
| Totten | 6,187.0 | -0.42 ± 0.50 | 9.55 ± 0.75 | -12.0 ± 2.8 |
| Conger Glenzer | 1,600.9 | 0.22 ± 0.29 | 1.37 ± 0.36 | -9.0 ± 1.6 |
| Tracy Tremenchus | 2,941.0 | 0.39 ± 0.12 | 1.00 ± 0.15 | -3.7 ± 0.7 |



| Ice shelf | Area (km$^2$) | Thinning rate (m/year) | Basal melt rate (m/year) | Thinning accel. (cm/year$^2$) |
|---|---|---|---|---|
| Shackleton | 26,927.9 | 0.52 ± 0.19 | 1.24 ± 0.27 | -0.7 ± 1.0 |
| West | 16,082.7 | -0.03 ± 0.18 | 0.76 ± 0.23 | 3.0 ± 1.0 |
| Publications | 1,563.3 | 1.08 ± 0.97 | 4.29 ± 1.06 | -8.3 ± 5.3 |
| Amery | 60,797.3 | -0.09 ± 0.20 | 0.40 ± 0.24 | 5.0 ± 1.1 |
| Prince Harald | 5,455.0 | -0.22 ± 0.24 | 0.66 ± 0.30 | 4.3 ± 1.3 |
| Baudouin | 33,129.2 | -0.03 ± 0.10 | 0.52 ± 0.11 | 2.2 ± 0.5 |
| Borchgrevink | 21,615.6 | 0.03 ± 0.12 | 0.28 ± 0.15 | -0.7 ± 0.7 |
| Lazarev | 8,571.6 | 0.07 ± 0.10 | 0.35 ± 0.12 | -0.7 ± 0.5 |
| Nivl | 7,321.5 | -0.25 ± 0.11 | 0.30 ± 0.13 | -0.4 ± 0.6 |
| Vigrid | 2,096.0 | -0.14 ± 0.12 | -0.03 ± 0.14 | 2.0 ± 0.6 |
| Fimbul | 40,947.7 | -0.17 ± 0.12 | 0.28 ± 0.16 | 2.2 ± 0.7 |
| Jelbart | 10,845.5 | -0.10 ± 0.15 | 0.33 ± 0.19 | 1.8 ± 0.8 |
| Atka | 1,993.7 | 0.21 ± 0.09 | 0.98 ± 0.14 | -1.0 ± 0.5 |
| Ekstrom | 6,870.8 | 0.11 ± 0.09 | 0.61 ± 0.13 | 2.3 ± 0.5 |
| Quar | 2,131.9 | -0.10 ± 0.13 | 0.59 ± 0.15 | 4.5 ± 0.7 |
| Riiser-Larsen | 42,913.1 | 0.09 ± 0.11 | 0.43 ± 0.12 | -1.1 ± 0.6 |
| Brunt Stancomb | 36,137.1 | 0.06 ± 0.28 | 0.09 ± 0.61 | -6.4 ± 1.5 |
| Filchner | 99,634.6 | -0.06 ± 0.12 | 0.14 ± 0.20 | 0.2 ± 0.6 |
| Amundsen | 55,790.5 | 2.61 ± 0.25 | 7.38 ± 1.08 | -8.3 ± 1.3 |
| Bellingshausen | 81,151.2 | 0.49 ± 0.18 | 2.13 ± 0.22 | -1.1 ± 1.0 |
| Antarctica | 1,541,700.0 | 0.19 ± 0.12 | 0.68 ± 0.18 | -1.3 ± 0.7 |

**Table 3: Antarctic ice shelf change mean values (1992—2017).** Thinning rate, basal melting and thinning acceleration (in meters of ice equivalent per year) are the 26-year mean values averaged over the respective ice shelf areas, excluding a 3-km buffer along the GLs and a 6-km buffer along the ice fronts. Area values shown (in squared km) refer to total ice-shelf area. Ice shelves smaller than 1 km$^2$ have been excluded due to the resolution limitation of the satellite altimeters.