# Peer review of "Widespread slowdown in thinning rates of West Antarctic Ice Shelves"

_EGUsphere, 2022_

## Referee Comment (RC1)

**Review of Paolo et al. 'Widespread slowdown in thinning rates of West Antarctic Ice Shelves'**

**Summary**

This study presents a new dataset of ice shelf thickness change derived from satellite radar altimetry from 1993 to 2017. The authors use this dataset to investigate the temporal evolution of ice shelf thickness patterns in the Amundsen and Bellingshausen Sea sectors over the course of this 26 year record showing that thinning rates have recently abated.

This paper confirms previous findings that thinning of ice shelves in the Amundsen Sea sector has slowed down since 2008 (e.g Adusumilli et al., 2020), and allows for further investigation of temporal fluctuations in ice shelves thinning and basal melt rates owing to methodological improvements leading to a higher temporal resolution. This paper is well-written and the methodology is clearly explained. However, I have some comments and suggestions that would need to be addressed before this paper can be published.

**Main comments**

- This study covers the years 1993 to 2017 but it might be worth considering extending this record further in time to cover the years 2018 to 2021 using the most recent CryoSat-2 baseline product (now in baseline E instead of baseline C), especially since Adusumilli et al. dataset (2020) covers the years 1994 to 2018. Adding more recent data would be a great addition to this paper and would add some interesting discussion on more recent changes in ice shelf thickness not published elsewhere. In addition, having an up-to-date dataset would be helpful to the scientific community.
- The methodological improvements implemented in this study are clearly explained in the manuscript. However there is no quantitative statements in the paper describing how those improvements have led to a better/more robust dataset compared to previous work. For instance, does the surface scattering correction effectively removes the height changes induced solely by changes in the scattering properties? What are the improvements afforded by the modified plane fitting procedure implemented here?
- The authors use the GEMB model to correct for firn air content. While the model is clearly explained in the paper, there is no validation of the model outputs. Has this model been used before in similar studies? Figure 3 shows the FAC volume change and SMB time-series from GEMB, GSFC FDM and IMAU FDM but this does not give enough information to the reader on how GEMB performs. It would be interesting to add a map of spatial differences between the three models as previous studies (e.g. Mottram et al. 2021) have shown that there are significant spatial differences between SMB models in Antarctica and while different models might agree on the total SMB, there might be some important biases regionally.
- The simple ice-ocean modelling experiment doesn't seem to bring much to the paper as the aim of this experiment is not very clear. In the abstract it is stated that this experiment will help test the resolution capability of the ice shelf thickness and basal melt rates datasets, but this is not discussed in the paper and there are no comparisons of observations and model results.

**Specific comments**

L79-80: How and at what stage of the processing chain do you bring the ascending/descending data points together?

L83-84: How many points are required to perform the bilinear/biquadratic fit? When do you pick one of the fits over the other?

L85: What range radii are you using?

L95: Please specify the range of your inversion cells size.

L100-104: Is your buffer size sufficient to account for grounding line migration in areas that have significantly evolved over the course of these 26 years?

L143-146: What is the magnitude of the correction based on correlations with the waveform parameters? Is there one of the waveform parameters that exhibits a higher correlation or do all three parameters need to be used together?

L208: What are the seven densification methods available in GEMB?

L251: Why using 5-month intervals? (It's stated later in the paper, but as it is first mentioned here, it'd be best to add this justification here)

L259: What's the average proportion of ice shelf area covered at each epoch before interpolation?

L260: How many random locations do you use to calculate the empirical covariances?

L271: 'improved' compared to?

L281: 'varying in time in the Amundsen Sea sector' and what about the Bellingshausen Sea sector? At L289, you construct a velocity product for both the Amundsen and the Bellingshausen Sea sectors so it's a bit confusing whether you are also using a time-variable dataset for the Bellingshausen Sea sector.

L282-284: There have been significant changes in ice flow in some basins of East Antarctica and in the Getz region as well. What's the time resolution of the ice velocity data used in the Amundsen Sea Sector? In recent years, annual velocity maps have been generated for the whole of the continent, can you use those annual maps to create a time-varying dataset for the whole of Antarctica for recent years?

L289-290: Why do you combine those two products? Are they complementary, do they use different satellite datasets or use a different methodology or cover different areas?

L391-392/L412-413: I suggest stating at the start of this section what variables you are computing for your comparison to Adusumilli et al. (2020) results as it will improve the clarity of this section.

L411-413: Can you give the total volume control that you use? I suggest also adding the total ice shelf extent area and number of ice shelves that your dataset covers earlier in the paper (these values are given in Table 3 but are worth adding the main text)?

L435-437: Can you quantify this 'good agreement'? This sub-section, while explaining how the different parameters have been calculated to allow a comparison to Adusumilli et al., is missing a paragraph on the differences/similarities found between the two datasets. For instance, remarking that accounting for thickness changes leads to a better match for this study would be an interesting point to make and needs to be expanded in the text.

L509: How many kilometres from the grounding line?

**Figure comments**

Figure 1: Can you add a reference to this statement on the GPS record in the caption?

Figure 6: Specify from what area this map shows.

Figure 7 (right panel): Can you label the ice shelves directly on the map rather than by order of magnitude given in the caption?

Figure 8: What does the black line represent on in the inset map?

Figure 9: What is the acquisition date of the grounding lines shown? Can you perhaps change the brightness of the green velocity vectors as they are a bit difficult to discern on the map?

Figure 11: As the mean values are plotted inside the error circles, it looks like the values refer to the error values and not the mean values. I suggest either increasing the font size to the size of the full circle (not just to fit the size of the inner error circle) or adding a mention on the figure directly (not just in the caption).

Figure 17: Can you add the same map of basal melt rates from your observations? Do the patterns of basal melt rates from observations in 1993 and 2017 look similar? It would be worth expanding on this in the main text as well to integrate the ice-ocean modelling experiment better with the rest of the paper.

Table 2: Can you add a column with the corresponding ice shelf areas over which the calculations have been made?

**Technical comments**

L233: missing word 'as a function of C'

L243: typo 'plateau'

L258: remove superscript '28'

L290: remove '(' at the end of the sentence

L348: 'comprises'

L415: remove 'where'

L483: 'reflects'

L506: '4.' Remove '.'

---

## Author Comment (AC1)

Dear Reviewer,

After carefully reading both reviewers' comments, we conclude that they did not find any major weaknesses or flaws with our manuscript. Our analysis of the comments can be summarized as:

**(i) adding four more years of data to the time series (R1)** – this is the most significant change suggested, and was only a suggestion for us to consider. We respond to this below.

**(ii) restructuring the m/s and further clarify some aspects of the methods (R2)** – this is straightforward for us to do, and we are happy to do this.

In response to (i) from R1. We agree that a longer, more complete, record is always desired. However, we explained in our response why it is not a simple matter, as it involves many steps and careful integration of models. We feel that the request is reasonable but should not be considered as a condition for publication. We feel it is appropriate to point out that this paper has now been through a few rounds of revision (with other journals before this), every one of them being fairly slow. We believe that this is why the data now seem dated. Regardless, we believe the dataset in its present form is of high value to the community and will be well received upon publication.

We also must provide you with some context to explain the conditions under which we are working. The lead author Fernando Paolo left academic research in 2020 and does not have the time nor the funding to do additional data analysis. We note that our dataset represents a significant improvement over the only other time-dependent dataset of ice shelf basal melt rates (spanning a similar time interval) currently available to the community (Adusumilli et al. 2020), and that the dataset presented here is already being used in large modeling initiatives (e.g. Estimating the Circulation and Climate of the Ocean – ECCO, see below) and published research (Nakayama et al. 2001, GRL; Greene et al. 2022, Nature).

The melt rates presented here have been used in producing ECCO's latest estimate, Version 4 Release 5, available at https://ecco.jpl.nasa.gov/drive/files/Version4/Release5

**REVIEWER #1 COMMENTS**

Review of Paolo et al. 'Widespread slowdown in thinning rates of West Antarctic Ice Shelves'

Summary

This study presents a new dataset of ice shelf thickness change derived from satellite radar altimetry from 1993 to 2017. The authors use this dataset to investigate the temporal evolution of ice shelf thickness patterns in the Amundsen and Bellingshausen Sea sectors over the course of this 26 year record showing that thinning rates have recently abated.

This paper confirms previous findings that thinning of ice shelves in the Amundsen Sea sector has slowed down since 2008 (e.g Adusumilli et al., 2020), and allows for further investigation of temporal fluctuations in ice shelves thinning and basal melt rates owing to methodological improvements leading to a higher temporal resolution. **This paper is well-written and the methodology is clearly explained**. However, I have some comments and suggestions that would need to be addressed before this paper can be published.

Thank you so much for taking the time to review our paper. We appreciate the insightful comments and suggestions you provided. Your constructive criticism is highly valued. We also appreciate that you consider the manuscript to be well written, as we took a lot of time to ensure this was the case.

Main comments

- This study covers the years 1993 to 2017 but it might be worth considering extending this record further in time to cover the years 2018 to 2021 using the most recent CryoSat-2 baseline product (now in baseline E instead of baseline C), especially since Adusumilli et al. dataset (2020) covers the years 1994 to 2018. Adding more recent data would be a great addition to this paper and would add some interesting discussion on more recent changes in ice shelf thickness not published elsewhere. In addition, having an up-to-date dataset would be helpful to the scientific community.

We first note that the only other published dataset of basal melt rates is composed of a temporally-static (i.e. time averaged) map at 500 m resolution for the CryoSat-2 period only, accompanied by a low-resolution (10 km) time series, published by Adusumilli et al. (2020). The high-resolution map is a great dataset but does not provide information on the year-2-year variability. Our dataset is the first, and only, 26-year-long quarterly dataset published at 3 km grid posting (with an effective resolution between 3 and 5 km). This makes our dataset unprecedented relative to the current state of the art. We include a figure at the end of this document that shows the capability of our time evolving product.

That said, we fully agree with the reviewer that extending the ice shelf melt record would be a great and logical next step. The challenge is that it is not just a simple matter of tagging on a few more data points; there are many ancillary data and models that we also need to run to extend the time series, including: SMB, firn and surface velocities. This would mean new model runs, blending velocity data sets with inconsistent time spans and spatial coverage, etc. For the new Cryosat-2 data we would need to recompute the surface scattering correction and assess any differences compared to the previous data. We would also need to re-estimate the spatial covariances for the optimal interpolation. All this would require significant work that is in the pipe for the next iteration of the dataset. All of this is sensible to do, however we should be required to do this for the paper being presented. Given that the lead author is no-longer working in academic research, such an effort would likely delay important publication and sharing of this work by a year or more.

- **The methodological improvements implemented in this study are clearly explained in the manuscript**. However there is no quantitative statements in the paper describing how those improvements have led to a better/more robust dataset compared to previous work. For instance, does the surface scattering correction effectively removes the height changes

induced solely by changes in the scattering properties? What are the improvements afforded by the modified plane fitting procedure implemented here?

The altimetry processing innovations we present are not the main focus of this paper, but they do represent a major methodological step change that makes previously invisible glaciological processes visible.

In the Quality Assessment section, we compare our results to Adusumilli et al. (2020), the only other existing large-scale time-variable (at low spatial resolution) melt rate product. We used a control-volume approach (entirely independent from the approach used in our analysis) to compare every ice-shelf estimate from both products. Our product shows significantly better statistics, with a variance of -0.2 ± 5.1 Gt/yr versus 1.4 ± 32.2 Gr/yr from Adusumilli et al. (2020), a 7-fold improvement. This is shown in Figure 10 in the manuscript, and we also include it at the end of this document.

We demonstrate the effect of our backscatter correction to the height records over Lake Vostok (Figure 1 in the manuscript), where GPS measurements are available confirming that it is a stable surface. Figure 1 shows clearly that our approach is successfully able to remove almost all backscatter-induced changes (compare with, for example, Zwally et al.'s (2015) Figure 7 [see below figure], where the authors of that paper were unable to fully correct for this effect over Lake Vostok). This is arguably the main limitation of radar altimetry over ice surfaces. As Paolo et al. (2016) showed, this effect can account for up to 80% of the height change signal over ice shelves. Here again we note that some state-of-the-art ice-shelf studies (e.g. Shepherd et al., 2018, Nature; Wouters et al. 2015, Science; Konrad et al. 2017, GRL) provide little to no information on how they have addressed this key problem over the ice shelves.

Our study

[Figure]

Figure 1: Multi-parameter radar scattering correction. (top) The different waveform parameters used to characterize the radar echo (where A is amplitude, N is the noise floor, and 'Range bins' are the discrete samples of the return signal). (bottom) Time series of individual point high measurements before and after applying the scattering correction. The example shows Lake Vostok where GPS records show no significant trend or variability in surface height.

Zwally et al, 2015

[Figure]

Zwally and others: Mass gains of the Antarctic ice sheet

[Figure]

Fig. 7. H(t) time series on Vostok Subglacial Lake. From ERS 1992–2003 with trend of +2.03 cm a⁻¹ after backscatter correction (red) and +2.18 cm a⁻¹ before backscatter correction (black). From ICESat 2003–08 with trend of +2.02 cm a⁻¹ (blue). The backscatter correction significantly reduces the amplitude of the seasonal variability in the ERS signal.

We also demonstrate that (a) there are clear spatial correlation scales for each quantity to be estimated (height, trend and acceleration; Figure 5 in the manuscript), and (b) there is an along-track pattern (artifacts) that originates from interpolating satellite data collected along tracks (Figure 6 in the manuscript). We show in Figure 6 a comparison for Ross Ice Shelf

between neglecting and accounting for these correlation lengths in the data fusion approach. We feel strongly that the result speaks for itself. If, for example, we zoom in on the Adusumilli et al. (2020) melt product, we will notice these artifacts.

- The authors use the GEMB model to correct for firn air content. While the model is clearly explained in the paper, there is no validation of the model outputs. Has this model been used before in similar studies? Figure 3 shows the FAC volume change and SMB time-series from GEMB, GSFC FDM and IMAU FDM but this does not give enough information to the reader on how GEMB performs. It would be interesting to add a map of spatial differences between the three models as previous studies (e.g. Mottram et al. 2021) have shown that there are significant spatial differences between SMB models in Antarctica and while different models might agree on the total SMB, there might be some important biases regionally.

This is a good point and we believe this can be satisfied by directing the reviewer to a soon to be published GEMB model description and validation paper. Since submission of our original manuscript, the GEMB team has written and submitted a paper to Geoscientific Model Development (https://egusphere.copernicus.org/preprints/2022/egusphere-2022-674/). The paper is all but accepted with the last status update being that "The referees are satisfied with your revised manuscript. They have left a range of minor comments and technical corrections. Once you have addressed these, please submit your revisions. I plan to review them (for oversight) but not to send your paper back out to the referees." The GEMB manuscript provides model algorithm details and an exhaustive comparison between GEMB and RACMO SMB and between GEMB and IMAU-FDM firn models. In addition GEMB is a publicly available module of the official Ice Sheet System Model (ISSM) release so others are welcome to use the model in their research or to investigate the code in further detail. We will heavily modify the description of the SMB and firn model accordingly (mostly by reducing text) and provide proper citation to the new GEMB manuscript in the revised manuscript.

- The simple ice-ocean modelling experiment doesn't seem to bring much to the paper as the aim of this experiment is not very clear. In the abstract it is stated that this experiment will help test the resolution capability of the ice shelf thickness and basal melt rates datasets, but this is not discussed in the paper and there are no comparisons of observations and model results.

We agree with the reviewer that our simplistic modeling exercise does not enhance the ice shelf thinning/melting results we present. Unfortunately, observations of ocean properties relevant to ice shelf processes are sparse, making any conclusions about changes in ocean forcing difficult to demonstrate. We have been careful not to come to any major conclusions based on sparse oceanographic data, and instead we have simply asked how melt rates are expected to change as the ice shelves of the Amundsen Sea Embayment thin themselves into cooler, shallower waters. So we deliberately used a simple "sandbox" experiment to help answer this important question. In addition, since we tailored our ice-shelf thickness and basal melt rate products to the modeling community (these data are already being incorporated into large modeling initiatives such as the NASA Estimating the Circulation and Climate of the Ocean – ECCO), we wanted to test the level of detail in modeled basal melt changes that was possible using our dataset.

The ten comments below are all details of the methods and we plan to include every one of these details in the new Methods section:

1. L79-80: How and at what stage of the processing chain do you bring the ascending/descending data points together?

   During the optimal interpolation. Ascending/Descending are treated as independent datasets until the end.

2. L83-84: How many points are required to perform the bilinear/biquadratic fit? When do you pick one of the fits over the other?

   biquadratic < 30pts < bilinear < 15 < mean value < 5 < NaN

3. L85: What range radii are you using?

   This is simply stating that this approach allows the use of different search radii if needed.

4. L95: Please specify the range of your inversion cells size.

   8-15 km

5. L100-104: Is your buffer size sufficient to account for grounding line migration in areas that have significantly evolved over the course of these 26 years?

   We did the best we could with the GL information we have available. The caveats with GL migration are stated in the manuscript.

6. L143-146: What is the magnitude of the correction based on correlations with the waveform parameters? Is there one of the waveform parameters that exhibits a higher correlation or do all three parameters need to be used together?

   The details of this can be found in the dedicated literature, e.g. Paolo et al. 2016, Nilsson et al. 2022 and 2016. A multivariate fit takes care of collinearity between the variables.

7. L208: What are the seven densification methods available in GEMB?

   We will now refer to the (soon-to-be-published) GEMB paper for the technical description of the model. See comment on GEMB.

8. L251: Why using 5-month intervals? (It's stated later in the paper, but as it is first mentioned here, it'd be best to add this justification here)

   Correct. It's stated later. We will add it here.

9. L259: What's the average proportion of ice shelf area covered at each epoch before interpolation? L260: How many random locations do you use to calculate the empirical covariances?

Coverage is homogeneous across ice shelves and fixed in time. We will add ice shelf surveyed areas in the revised manuscript. In the latest iteration of the dataset we used the full Ross Ice Shelf area worth of data (about 1/3 of all Antarctic ice shelf area). We will clarify.

10. L271: 'improved' compared to?

To previous work? (Figure 10 in the manuscript compares our results with previous work. We also add the figure at the end of this document)

L281: 'varying in time in the Amundsen Sea sector' and what about the Bellingshausen Sea sector? At L289, you construct a velocity product for both the Amundsen and the Bellingshausen Sea sectors so it's a bit confusing whether you are also using a time-variable dataset for the Bellingshausen Sea sector.

We will clarify that we use time-varying velocity in the Amundsen and the Bellingshausen Sea sectors.

L282-284: There have been significant changes in ice flow in some basins of East Antarctica and in the Getz region as well. What's the time resolution of the ice velocity data used in the Amundsen Sea Sector? In recent years, annual velocity maps have been generated for the whole of the continent, can you use those annual maps to create a time-varying dataset for the whole of Antarctica for recent years?

This is a totally reasonable question. We do not yet use time-varying velocity in any region other than the Amundsen and the Bellingshausen Sea sectors. That said, Gardner et al. (2018) show that, for the period with satellite data, observed changes in surface velocity outside of these two regions are very small. We do not expect that the inclusion of more velocity data would change the results presented. If anything, the noise introduced by reduced time-averaging of velocity observations in areas with little to no changes in velocity could introduce more noise into estimates of melt rates. For these reasons we did not include time-varying velocities for other regions.

L289-290: Why do you combine those two products? Are they complementary, do they use different satellite datasets or use a different methodology or cover different areas?

Exactly, we synthesize multiple datasets to provide the best possible estimate of glacier surface velocity in a highly dynamic region of the ice sheet. The dataset of Mouginot et al. provides a long-localized record for the period 1973 to 2013 while the ITS_LIVE dataset provides pan Antarctic coverage for the period 2014 to 2018 (at time of submission). We will include this in Methods.

L391-392/L412-413: I suggest stating at the start of this section what variables you are computing for your comparison to Adusumilli et al. (2020) results as it will improve the clarity of this section.

Good suggestion, we will include this.

 Can you give the total volume control that you use? I suggest also adding the total ice shelf extent area and number of ice shelves that your dataset covers earlier in the paper (these values are given in Table 3 but are worth adding the main text)?

Good point, we will make sure to do this in the revision.

 Can you quantify this 'good agreement'? This sub-section, while explaining how the different parameters have been calculated to allow a comparison to Adusumilli et al., is missing a paragraph on the differences/similarities found between the two datasets. For instance, remarking that accounting for thickness changes leads to a better match for this study would be an interesting point to make and needs to be expanded in the text.

Thanks for the suggestion, we will include this in the revision.

 How many kilometres from the grounding line?

Seems like a valuable thing to include. We will address this in the revision.

Figure comments

1. Figure 1: Can you add a reference to this statement on the GPS record in the caption? Figure 6: Specify from what area this map shows.

   Thank you for catching that. we will add that in our resubmission

2. Figure 7 (right panel): Can you label the ice shelves directly on the map rather than by order of magnitude given in the caption?

   Yes, that makes more sense. We will do this in the revision.

3. Figure 8: What does the black line represent on in the inset map?

   Thanks for noticing that, it's a relic from an earlier iteration of the paper. We will remove that in the revision.

4. Figure 9: What is the acquisition date of the grounding lines shown? Can you perhaps change the brightness of the green velocity vectors as they are a bit difficult to discern on the map?

   We will include the date of the grounding line and modify the vectors in the revision.

5. Figure 11: As the mean values are plotted inside the error circles, it looks like the values refer to the error values and not the mean values. I suggest either increasing the font size to the size of the full circle (not just to fit the size of the inner error circle) or adding a mention on the figure directly (not just in the caption).

   If we increase the font size the darker outer band will make it difficult to see the numbers (see WIL for example).. We feel the best solution is to mention on figure directly. We will do this in the revised manuscript.

6. Figure 17: Can you add the same map of basal melt rates from your observations? Do the patterns of basal melt rates from observations in 1993 and 2017 look similar? It would be worth expanding on this in the main text as well to integrate the ice-ocean modelling experiment better with the rest of the paper.

We see your point regarding the observations but in this case that wouldn't make a lot of sense as the figure is showing a theoretical model experiment with fixed ocean condition and modified ice shelf geometry. Here we are simply trying to demonstrate that changes in ice shelf geometry alone result in changes in basal melt rates, even in the absence of changes in ocean state.

7. Table 2: Can you add a column with the corresponding ice shelf areas over which the calculations have been made?

Good suggestion, we'll add this in the resubmission.

**Technical comments**

All five of these typos will be fixed, thank you for catching them.

1. L233: missing word 'as a function of C' L243: typo 'plateau'
2. L258: remove superscript '28'
3. L290: remove '(' at the end of the sentence L348: 'comprises'
4. L415: remove 'where' L483: 'reflects'
5. L506: '4.' Remove '.'

**Additional References**

Greene, C.A., Gardner, A.S., Schlegel, NJ. *et al.* Antarctic calving loss rivals ice-shelf thinning. *Nature* 609, 948–953 (2022). https://doi.org/10.1038/s41586-022-05037-w

Nakayama Y., C.A. Greene, F.S. Paolo, et al. (2021), Antarctic Slope Current Modulates Ocean Heat Intrusions Towards Totten Glacier, Geophysical Research Letters, https://doi.org/10.1029/2021GL094149

Nilsson, J., Gardner, A. S., and Paolo, F. S.: Elevation change of the Antarctic Ice Sheet: 1985 to 2020, Earth Syst. Sci. Data, 14, 3573–3598 (2022), https://doi.org/10.5194/essd-14-3573-2022

Nilsson, J., Gardner, A., Sandberg Sørensen, L., and Forsberg, R.: Improved retrieval of land ice topography from CryoSat-2 data and its impact for volume-change estimation of the Greenland Ice Sheet, The Cryosphere, 10, 2953–2969 (2016), https://doi.org/10.5194/tc-10-2953-2016

Holland, P. W. and Welsch, R. E.: Robust regression using iteratively reweighted least-squares, Commun. Stat. – Theory Methods, 6, 813–827 (1977), https://doi.org/10.1080/03610927708827533

Zwally, H., Li, J., Robbins, J., Saba, J., Yi, D., & Brenner, A. Mass gains of the Antarctic ice sheet exceed losses. Journal of Glaciology, 61, 1019-1036 (2015), http://doi.org/10.3189/2015JoG15J071

Wouters, Bert & Martín-Español, Alba & Helm, Veit & Flament, Thomas & Wessem, J. M. & Ligtenberg, Stefan & Van den Broeke, Michiel & Bamber, Jonathan, Dynamic thinning of glaciers on the Southern Antarctic Peninsula. Science. 348. 899-903 (2015), https://doi.org/10.1126/science.aaa5727

Konrad, H., L. Gilbert, S. L. Cornford, A. Payne, A. Hogg, A. Muir, and A. Shepherd (2017), Uneven onset and pace of ice-dynamical imbalance in the Amundsen Sea Embayment, West Antarctica, Geophys. Res. Lett., 44, 910–918 (2017), https://doi.org/10.1002/2016GL070733

Paolo, F.S., Fricker H.A., Padman L., Constructing improved decadal records of Antarctic ice shelf height change from multiple satellite radar altimeters, Remote Sensing of Environment, Volume 177, 192-205 (2016), https://doi.org/10.1016/j.rse.2016.01.026

Paolo, F.S., Padman, L., Fricker, H.A. et al. Response of Pacific-sector Antarctic ice shelves to the El Niño/Southern Oscillation. Nature Geosci 11, 121–126 (2018). https://doi.org/10.1038/s41561-017-0033-0

[Figure]

**Figure 1: Example of our high-resolution time-variable basal melt rates.** Melt anomalies (in meters of ice per year) with respect to the 26-year mean (1992-2017), where red is faster-than-normal melting and blue is slower-than-normal melting. Note the rapid change in basal melt patterns at interannual timescales. Also note that coherent melt structures form in different parts of the ice shelf at different epochs. Example is the Filchner-Ronne ice shelf in Antarctica. This temporal mapping of basal melt rates is only possible due to a consistent 3-km grid posting from 1992 to 2017 at 3-month timesteps. This is also what allowed us to estimate with high confidence the second derivative (acceleration/deceleration) of the thickness and melt records.

[Figure]

**Figure 2: Comparison of melt estimates from this study with previous work.**
Comparison of (left) Adusumilli et al. (2020) and (right) this study's ice shelf basal melt estimates against a control-volume calculation of ice shelf mass change (gray line). The control volume is based on the input and output fluxes across the grounding line and ice front, i.e. ice loss due to anomalies in basal melt. For this comparison, we only considered grid cells that are at least 90% hydrostatically compensated (near fully floating).

---

## Author Comment (AC2)

Dear Reviewer,

After carefully reading both reviewers' comments, we conclude that they did not find any major weaknesses or flaws with our manuscript. Our analysis of the comments can be summarized as:

**(i) adding four more years of data to the time series (R1)** – this is the most significant change suggested, and was only a suggestion for us to consider. We respond to this below.

**(ii) restructuring the m/s and further clarify some aspects of the methods (R2)** – this is straightforward for us to do, and we are happy to do this.

In response to (i) from R1. We agree that a longer, more complete, record is always desired. However, we explained in our response why it is not a simple matter, as it involves many steps and careful integration of models. We feel that the request is reasonable but should not be considered as a condition for publication. We feel it is appropriate to point out that this paper has now been through a few rounds of revision (with other journals before this), every one of them being fairly slow. We believe that this is why the data now seem dated. Regardless, we believe the dataset in its present form is of high value to the community and will be well received upon publication.

We also must provide you with some context to explain the conditions under which we are working. The lead author Fernando Paolo left academic research in 2020 and does not have the time nor the funding to do additional data analysis. We note that our dataset represents a significant improvement over the only other time-dependent dataset of ice shelf basal melt rates (spanning a similar time interval) currently available to the community (Adusumilli et al. 2020), and that the dataset presented here is already being used in large modeling initiatives (e.g. Estimating the Circulation and Climate of the Ocean – ECCO, see below) and published research (Nakayama et al. 2001, GRL; Greene et al. 2022, Nature).

The melt rates presented here have been used in producing ECCO's latest estimate, Version 4 Release 5, available at https://ecco.jpl.nasa.gov/drive/files/Version4/Release5

**REVIEWER #2 COMMENTS**

SUMMARY

This research by Paolo and colleagues analyzes 26 years of data from satellite observations of Antarctic ice shelf thickness based on a novel data fusion approach, advanced satellite-derived velocities, and a new surface mass balance model (GEMB). The changes in ice shelf thickness are subsequently related to (changes in) flow and basal melt rates. The study found a pattern of overall thinning around Antarctica, with a slowdown in thinning starting around 2008 (similar to earlier research of for example Adusumilli et al., 2020). The researchers attribute this slowdown to changes in external ocean forcing and potential

negative feedback effects of i) accelerated grounded ice flow on ice shelf thinning rates and ii) thinning and melt rates.

We want to preface our response with a big thank you. We know that you are likely very busy and that taking the time to provide a thoughtful, in-depth, review is not easy. We are grateful for your time and input. Your constructive criticism is highly valued. We can see that a restructuring of parts of the manuscript would be very valuable to add clarity, so thank you for bringing that to our attention. We do disagree with other comments but we hope that we have provided sufficient context for you to understand our point of view. We see several of the comments as good suggestions that should not be considered a condition for publication.

MAJOR COMMENTS

This research touches upon an important topic, is original with several novelties and provides new understanding of the melting of ice shelves. Therefore, I think the manuscript ultimately warrants publication in the Cryosphere if it manages to tackle some of the major and specific comments listed below:

   -[1] Restructuring: although the paper is overall well written, I think the structure often is complicated to follow with data and methods that are often intermingled and without a clear distinction of results and discussion.

We understand that the current structure of the manuscript might not be optimal. There are mainly two reasons for this. First, our manuscript indeed presents a combination of new data, novel methods, and new findings. It is challenging to maintain a balance throughout the manuscript as some topics are more technical in nature and, therefore, require more extensive descriptions. Second, we tried to follow the somewhat rigid format of the TCD regarding the different sections.

     ○[2] I think the paper would benefit from a more extensive Data section, where next to the Radar data, also the velocity data, GEMB model data and ECCO model data are described. These data sets now are gradually introduced throughout the manuscript and it is therefore sometimes difficult to keep the overview.

We agree with the reviewer that the manuscript would benefit from a more traditional manuscript layout.  As the main issue seems to be an intermingled data/methods section, for the revised manuscript we will recognize the material into the IMRAD format of:

a)  Data
b)  Methods
c)  Results
d)  Discussion

   ○ [3] I think the paper would benefit from a more extensive discussion that actually zooms out, puts the results in a wider context (e.g. relative to the state-of-art) and reflects on the implications. Currently this reflection is very limited

It may be a bit surprising but there is actually only one other high-resolution published map of basal melt rates (Adusumilli et al. 2020). That dataset represents the state-of-the-art and

we compare our work extensively to that. We interpreted the findings and discussed the wider implications the best we could within the scope of the manuscript, which is mainly to present a new dataset of ice shelf thickness change and basel melt rates. Accordingly, we must describe in detail the methodology to obtain such estimates. And because the methods and auxiliary datasets can be quite complex, there is a justifiable unbalance in the amount of text dedicated to "Data and Methods" versus "Results and Discussion".

- [4] Balance of detail: as a reader (and not a core altimetry expert) I was often missing the necessary details to understand the steps in the data processing and reproduce them. Therefore, I think it is key to provide much more detail on the processing methodology at several locations (especially for the steps that are at the core of the altimetry processing). At the same time, I had the impression that other parts (especially the GEMB model section) was very extensive and contained details that are less relevant for this paper (and can be looked up in the GEMB paper). I would therefore limit the GEM section to the core explanation and limit the description here to the steps that are specific to the paper (i.e. calibration of the densification parameters)

This specific comment is in contradiction with the other Reviewer that stated that the: "The methodological improvements implemented in this study are clearly explained in the manuscript". This is not surprising as readers/reviewers come from a wide range of backgrounds and will not always have the same perception of the paper.

That said, there are many standard procedures that one must follow in using altimetry data for ice-sheet/ice-shelf studies. There is a wealth of literature specific to the use of radar and laser altimeters over Antarctica, Greenland, and mountain glaciers, with detailed descriptions of these measurements and common processing approaches (e.g. Arthern et al., 2001; Bamber 1994; Borsa et al, 2014; Brenner et al, 1983; Brenner et al., 2007; Brockley et al., 2017; Davis 1993; Davis & Ferguson, 2004; Flament & Rémy, 2012; Helm et al., 2014; Hurkmans et al., 2012; Khvorostovsky 2012; Lacroix et al. 2009; Legresy and Remy, 1997; McMillan et al., 2014; Nilsson et al., 2016; Paolo, et al., 2016; Roemer et al, 2007; Schröder et al., 2017; Schröder et al., 2019; Simonsen et al.,2017; Wingham er al. 1986, 1998, 2006, 2009; Zwally et al., 2005, 2012, 2015, 2021). We cite the relevant literature throughout the manuscript. We follow the standard practice of describing in more detail only the novel aspects of the altimetry processing that we introduce and cite literature to cover those methods that have been previously described.

Regarding the GEMB section. At the time of submission, GEMB did not have a stand-alone publication which is why we provided such an extensive description of the model. Modeling of SMB and firn air content is of first order importance for inverting basal melt rates from satellite altimetry. That said there is now a GEMB paper that is near acceptance in Geophysical Model Developments that we can now cite (https://egusphere.copernicus.org/preprints/2022/egusphere-2022-674/). We will modify this section in the revised manuscript and cite the new paper appropriately (also see response to other reviewer)

We also want to point out that the analysis and description of the altimetry data and firn modeling matches or exceeds that of previous ice shelf studies (e.g. Paolo et al. 2015; Paolo

et al. 2018; Adusumilli et al. 2021; Smith et al. 2020; Shepherd et al. 2018, and many others).

- [5] Reproducibility: due to the lack of detail on several processing steps (or an condensed explanation for their motivation) I think it is basically impossible to reproduce the method. I applaud the effort of the authors to make code available, but with this type of code a user cannot reproduce the paper. It is a collection of methods and example notebooks that are not related to the paper. I think it is key that the code is provided as such that the reader can at least see which functions are used for which steps etc. Currently this code does not help reproducibility. I would therefore advice to either provide a notebook that runs through a complete workflow (preferential) and/or clearly indicate how the different blocks of code were used in which steps.

There might be a confusion regarding reproducibility of our work. Our data processing includes several essential steps, some of which are standard practice in using altimetry data for glaciological studies and, therefore, are exhaustively described in the dedicated literature (see, for example, Nilsson et al. 2022 and 2016); and some that were developed specifically for this work that we describe in detail (as acknowledged by the other Reviewer). We also point out that some introduced techniques, such as the optimal interpolation accounting for correlated errors, while novel in this field, these methods are widely used in other areas. So we provide the original references instead of reproducing an unnecessary lengthy description. Much of our analysis builds heavily on previous work.

The many processing steps and large-scale analysis with 26 years of satellite altimetry data, satellite velocities, surface mass balance modeling, and auxiliary geophysical fields, were performed with the aid of a supercomputer (HALO at JPL). It would be virtually impossible to provide code that would run on a personal computer. Nonetheless, all of our data and code have been made publicly available, increasing transparency and reducing barriers to future efforts. To our knowledge, our work is the first large-scale altimetry effort looking at ice shelves (or ice sheets for that matter) that has released all of their code and made all of the data publicly available. We point to another analysis of ice shelf melt rates recently published in Nature (Adusumilli et al. 2021) for which they satisfied the reproducibility by simply providing the statement that "The Matlab, Python and shell scripts used for the analyses described in this study can be obtained from the corresponding author upon reasonable request.". We feel we have gone far beyond the minimum requirement for reproducibility and have provided the glaciological community with a well documented Python library that is able to reproduce the entirety of the analysis presented in this manuscript (https://github.com/nasa-jpl/captoolkit) and has already been used in other published works, despite being developed for the study presented here (e.g. Smith et al., 2019; Alley et al. 2021; Nilsson et al., 2022).

We applaud the reviewer's concern and attention to reproducibility (something that we are also passionate about) but we feel that we have gone far beyond the minimum acceptable requirement and that requesting notebook style workflow that can run on other systems is a nice to have but should not negatively impact the evaluation of this work.

- [6] Dynamic vs melting processes: the authors show two processes that could explain the slowdown (i.e. increased ice flow and thinning into cooler water), but fail to provide a

consistent overview of the relative role of both (although they have the data to do so). Currently, they show individual examples for 4 ice shelves (Table 2) where they quantify the role of dynamic thinning vs. melting,. I suggest that they do this quantification for all ice shelves (in Table 3) and also add the role of dynamic thinning in all plots of time series to allow the reader to better understand the importance of both processes.

There are two important things to keep in mind. The main objective of this manuscript is to present a dataset of ice shelf thickness and basal melt rates. This is, however, accompanied by an extensive analysis that highlights some novel findings (perhaps we could adjust the title to better reflect this). The dynamic component (i.e. ice velocity) is needed in order to derive the basal melting from the total thickness change. While we make an effort to obtain the best available ice velocity estimates for the ice shelves in question, further analysis into the ice shelf dynamics is beyond the scope of this manuscript (which is already quite extensive).

Another key point, as explicit in the title, is the focus on "West Antarctic Ice Shelves". There are two reasons for this. First, this is the region where the thickness change signal is highest and, second, this is where we have the most complete time-dependent velocity data. So we could not make a similar temporal analysis for the ice shelf dynamics (and consequently basal melt rates) for all ice shelves. As stated in the manuscript, we use a mean velocity field for all ice shelves other than Pine Island, Thwaites, Crosson and Dotson (the focus of our paper).

**SPECIFIC COMMENTS**

L50: I think it would be good to have a separate data section where next to the Radar data, also the velocity data, GEMB model data and ECCO model data are described

This is a good suggestion and it would certainly improve the readability of the paper. Please see response to the second major comment.

L80-85: this methodological step might need a bit more extensive explanation as it is currently very condensed and therefore not easy to interpret unambiguously. It would also be good practice to clearly show each of these steps in the available code (reproducibility). Now the reader has to guess what happens and also does not have the code to better understand what happens.

The code and its documentation is available here (https://github.com/nasa-jpl/captoolkit) and is already being used by others to process satellite altimetry data (e.g. Smith et al., 2019; Alley et al. 2021; Nilsson et al., 2022).I think what the reviewer is asking for is a notebook that could be run by the user to execute each step of the processing. As mentioned in the response to the 5th major point, this is not easily doable as the code was designed to run on large datasets with the aid of supercomputers. Refactoring the code to provide localized examples that can be run on personal computers, though valuable as an explanatory tool, is outside of the scope of this project.

L92-96: again this methodological step might need a bit more extensive explanation as it is currently very condensed and therefore not easy to interpret unambiguously and/or to understand the motivation. It would also be good practice to clearly show each of these

steps in the available code (reproducibility). Now the reader has to guess what happens and also does not have the code to better understand what happens.

Please see responses to major comment 4.

L112 (but also later throughout the paper): when using text citations the brackets should be placed differently

Thank you for catching this. We will fix this in the revised manuscript

L128-130 "Global MDT … ice shelves": not clear if this is in general or in this study?

This is a general problem with Global MDT. We will clarify this in the revision.

L131-136: again this methodological step might need a bit more extensive explanation as it is currently very condensed and therefore not easy to interpret unambiguously and/or to understand the motivation. It would also be good practice to clearly show each of these steps in the available code (reproducibility). Now the reader has to guess what happens and also does not have the code to better understand what happens.

Again, we want to emphasize that the code is available but likely not as user friendly as the reviewer is hoping for. Please see responses to major comment 4.

L144-147: again this methodological step might need a bit more extensive explanation as it is currently very condensed and therefore not easy to interpret unambiguously and/or to understand the motivation. For example it is not clear what robust multi-variate regression is (ref?), if it is done for every pixel separately?

We will clarify, and point to a technical reference (Holland and Welsch, 1977).

L149-217: I think this section can be strongly reduced to keep only a condensed overview of GEMB (while refering to the GEMB main paper). Many of these methodological details are not relevant for this study (in contrast to other locations where the text is extremely condense on things that are important)

Please see response to major comment 4.

L223: what is the typical depth of 550 kg/m3? If it is not extremely deep, why would you need a spinup of

The 550 km/m3 depth varies from 5 to 50 meters in depth, depending on its location (for more please see the GEMB manuscript Gardner et al., 2023 Figure 4).  Spinup to a historical climate state is necessary in order to remove spurious trends in densification at both the 550 and 830 density horizon.  In this way, modeled forward densification response is ensured to be resulting from the climatological forcing of the simulation. This model spinup procedure is the standard in the literature (i.e. to produce other state of the art firn products like RACMO-FDM and GSFC-FDM).

7750 years as the firn will be mostly dependent what happens in the last years of the spinup period.

Trends in firn air content (FAC) are highly sensitive to the spin up period. The model must be run to equilibrium otherwise there will be residual/erroneous trends in FAC that will adversely impact estimates of glacier mass change. Through our analysis we determined that a spinup < 7750 years will produce model results with erroneous trends in FAC.

L257: 28 seems reference is in wrong referencing system

Good catch! We will fix it. Thanks

L260: what are 5 month bins for every 3 months?

This is a sliding window. At every 3 months time steps we aggregate (bin) 5 months worth of data, so each time window overlaps by one month on both ends.

L259- 269: again these methodological steps might need a bit more extensive explanation as it is currently very condensed and therefore not easy to interpret unambiguously and/or to understand the motivation. It would also be good practice to clearly show each of these steps in the available code (reproducibility). Now the reader has to guess what happens and also does not have the code to better understand what happens.

We can add further clarification to some of these concepts.

L276: for readability it would help if the actual terms in the equation are repeated

We will replace the acronyms with the respective equation terms.

L292-301: I do not understand the processing of the velocity data. Again these methodological steps might need a bit more extensive explanation as it is currently very condensed and therefore not easy to interpret unambiguously and/or to understand the motivation. It would also be good practice to clearly show each of these steps in the available code (reproducibility). Now the reader has to guess what happens and also does not have the code to better understand what happens.

We agree that the description of how the velocity data is merged is quite dense but the description is explicit and complete. Both datasets are freely available and all processing steps are explicitly defined, and therefore fully reproducible. A notebook would be a great addition to reduce barriers for others to replicate the work presented but, again, we do not think that it should be a necessary condition for publication. We will revisit this paragraph during revision to see if you can improve the overall readability.

L305: what are insignificant changes? How insignificant are they?

Indistinguishable from noise given the magnitude of the uncertainties in the data.

L312 "mean rate of thickness change from both ends of the trend fit" Is it just first vs last value or are annual mean or so used?

Correct. First vs last value (which is a conservative estimation in the presence of nonlinear trends).

L317: what is k? I guess the iterator over n?

Correct. We follow the convention of using $i,j$ for spatial and $k$ for temporal iterators.

Section 2.8: I think the uncertainty quantification needs much better explanation. Currently many of the uncertainty terms fall out of the sky and their motivation and derivation is unclear. As such it is difficult for the reader to assess what these uncertainty terms are and/or what they mean.

We present an extensive description of the uncertainties compared to any previous ice shelf melt work. The estimation of our uncertainties is in fact quite standard. It is mostly based on quadratic propagation of the reported errors (e.g. from model outputs and velocities) and calculated standard deviations, adjusted for the degrees of freedom in question.

Section 2.9 seems a section that is not properly placed. It is self-contained and contains results before even the main results are shown. I propose to move this section to supplementary material or appendix to allow the paper to focus on the main story

We can rearrange the structure of the manuscript as specified above.

L438: I think it is good practice to separate results and discussion so the discussion can really reflect on the outcome of the results

Agreed. We were following the TCD suggested layout. We can rearrange the structure.

L453-456: this is a methodological description and should not be part of the results

We can try moving this paragraph when rearranging the structure.

L484-486: this is a methodological description and should not be part of the results

This information is providing context for the "results and discussion". With the new structure (see comment above) all this will be better separated.

L490-492: this is interesting and is probably the key of the paper. I therefore suggest that the separation of thinning factors is quantified for all ice shelves and not only done in an anecdotal way for only 4 ice shelves.

We addressed this in detail above. These ice shelves are the focus of the analysis.

L496-497: I think this probably on overstatement as it might be read that it counts for all ice shelves. This might be true for Dotson, but is definitely not true for all ice shelves.

Again, every statement is referring to the ices shelves highlighted in this study.

L498-502: I think the context and perspective is important here. By just showing the changes in time-variable melt, the reader only sees one part of the puzzle. Therefore, it is important to simultaneously show the (time series of) dynamic contribution so the reader can see both terms in perspective.

The main objective of this manuscript is to introduce a dataset of ice shelf thickness and basal melt rates (perhaps we should modify the title to better reflect this). We however went a step further and also presented novel results. A more in depth analysis of the ice shelf dynamics is beyond the scope of the manuscript (we do quantify and state the dynamic contributions for the ice shelves in question). Moreover, to oceanographers and modelers using our melt rate product what matters is meltwater production.

L517-520: this is a methodological description and should not be part of the results

This will hopefully be addressed with the restructuring.

L526-528: please provide references for these notions

We will add the references.

L533-534: I think it is key for this paper that the relative contribution of this tandem should be quantified in order to understand the relative contribution of both processes

This is an attempt to offer a plausible explanation for what the data show. This paper is not about an in-depth study on ice shelf dynamics. (and it's already fairly extensive)

L539-542: again I think it is not good enough to hypothesize and hand-waive at this feedback mechanism. I think the authors have the data to quantify the relative contribution of both processes to asses their relative contribution

Again, this paper is not an in-depth investigation on ice shelf dynamics. It is a derivation of a dataset of ice shelf thickness and basal melt rates. We think it is perfectly acceptable to offer a plausible explanation consistent with our current knowledge (and the physics) of the ice shelf system.

Conclusion: this section is written in a complete different style (all we sentences). It would benefit from a more general conclusion style

We are not sure what "a more general conclusion style" means.

L556-561: I do not agree that these feedback mechanisms are not included. An ice sheet model advects ice (and hence thickness) and ice shelf draft plays a role in the melt parametrizations. Therefore, I do not see why the models should not account for these feedback processes.

Point taken. We will revise this statement. Also, if the physics of these feedback mechanisms is well understood, our explanation above is well-founded.

FIGURES

Figure 1: I don't think this figure add a lot to the understanding of the methodology (especially not a)) and provides an unbalanced view of the methodology (e.g. why only a figure of this step) as it is very anecdotal. I think it would be much more insightful to have a flowchart (or equivalent) of the complete processing steps instead of an anecdotal figure.

Lake Vostok is a reference site for testing altimetry measurements over ice surfaces. Previous studies have used this same location to compute records of height change. This step (the backscattering correction) is arguably the single most important correction applied to radar altimetry measurements over icy surfaces. The end product (the time series of height change) over Lake Vostok must show almost no long-term change in height. This is what we are showing.

Figure 3+4: I would also opt to remove this figure. They show GEMB results but are not necessary relevant for the key message of this paper.

We can remove these figures as we are shortening the GEMB section (see above comment), and delegating most of the technical description of the GEMB model to the respective (about to be published) paper.

Figure 5: it would be insightfull to also plot the data based on which these curves are estimated to give the reader an impression on how representative these estimates of decorrelation are

This work is based on satellite altimetry data, which means data coverage is extensive and dense, with data density increasing with latitude, characteristic of all satellite data.

Figure 6: Scalebar? Location?

This is Ross Ice Shelf. We will add that to the caption.

Figure 7: why is only half of Antarctica shown?

Because, as the title of the manuscript says, our focus is "West Antarctic Ice Shelves". Also, just a matter of practicality. Having the full continent would not facilitate visualization as the overall figure will be reduced and the ice shelves will become too small to see.

Figure 8: I would also opt to remove this figure. It adds very little to the understanding of the paper and it would be more insightfull to show figures that actually show the processing steps

This figure is key to show the domain of the modeling exercise, the four ice shelves we focus on.

Figure 10: what explains the enormous changes for Nivl and Lazarev between this study and Adusumulli?

We haven't looked into individual ice shelves outside the ones we focus on in this study.

Figure 11: nice figure!

Thanks! These figures take a lot of work.

Figure 14: It would be good to indicate in these figures how much can be explained by dynamic processes and how much by basal melt changes

Figure 16: for perspective it would be good to also plot the time series of the dynamic component to allow to put both processes into context

The main point of these figures is to show precisely the total thickness/meltwater signal. We quantify the dynamics in order to derive the basal melting. As we explained above, we only have time-variable velocities (i.e. dynamics) for Pine Island, Thwaites, Crosson and Dotson. All other ice shelves use a time-average velocity field (as done in previous work).

**Additional References:**

Greene, C.A., Gardner, A.S., Schlegel, NJ. *et al.* Antarctic calving loss rivals ice-shelf thinning. *Nature* 609, 948–953 (2022). https://doi.org/10.1038/s41586-022-05037-w

Nakayama Y., C.A. Greene, F.S. Paolo, et al. (2021), Antarctic Slope Current Modulates Ocean Heat Intrusions Towards Totten Glacier, Geophysical Research Letters, https://doi.org/10.1029/2021GL094149

Nilsson, J., Gardner, A. S., and Paolo, F. S.: Elevation change of the Antarctic Ice Sheet: 1985 to 2020, Earth Syst. Sci. Data, 14, 3573–3598 (2022), https://doi.org/10.5194/essd-14-3573-2022

Nilsson, J., Gardner, A., Sandberg Sørensen, L., and Forsberg, R.: Improved retrieval of land ice topography from CryoSat-2 data and its impact for volume-change estimation of the Greenland Ice Sheet, The Cryosphere, 10, 2953–2969 (2016), https://doi.org/10.5194/tc-10-2953-2016

Holland, P. W. and Welsch, R. E.: Robust regression using iteratively reweighted least-squares, Commun. Stat. – Theory Methods, 6, 813–827 (1977), https://doi.org/10.1080/03610927708827533

Zwally, H., Li, J., Robbins, J., Saba, J., Yi, D., & Brenner, A. Mass gains of the Antarctic ice sheet exceed losses. Journal of Glaciology, 61, 1019-1036 (2015), http://doi.org/10.3189/2015JoG15J071

Wouters, Bert & Martín-Español, Alba & Helm, Veit & Flament, Thomas & Wessem, J. M. & Ligtenberg, Stefan & Van den Broeke, Michiel & Bamber, Jonathan, Dynamic thinning of glaciers on the Southern Antarctic Peninsula. Science. 348. 899-903 (2015), https://doi.org/10.1126/science.aaa5727

Konrad, H., L. Gilbert, S. L. Cornford, A. Payne, A. Hogg, A. Muir, and A. Shepherd (2017), Uneven onset and pace of ice-dynamical imbalance in the Amundsen Sea Embayment, West Antarctica, Geophys. Res. Lett., 44, 910–918 (2017), https://doi.org/10.1002/2016GL070733

Paolo, F.S., Fricker H.A., Padman L., Constructing improved decadal records of Antarctic ice shelf height change from multiple satellite radar altimeters, Remote Sensing of Environment, Volume 177, 192-205 (2016), https://doi.org/10.1016/j.rse.2016.01.026

Paolo, F.S., Padman, L., Fricker, H.A. et al. Response of Pacific-sector Antarctic ice shelves to the El Niño/Southern Oscillation. Nature Geosci 11, 121–126 (2018). https://doi.org/10.1038/s41561-017-0033-0

---

## Author Response (AR1)

Dear Editor and Reviewers,

We would like to thank you for your constructive feedback on our manuscript. We appreciate your interest in our work and the time you have put into revising our paper.

We have addressed, explicitly in the text of the manuscript, all reviewer's questions and suggestions regarding expanding data and methods. We have also restructured the entire manuscript to better present: Introduction, Data, Methods, Results, Discussion, and Conclusion. We have added several new subsections, expanded existing ones, and condensed others (as suggested); and have also added additional text to further clarify our findings and justify our experiments.

In addition, we have included a supplementary file (`RECIPE.md`) with the full recipe (the various processing steps and code) we have used to construct our pan-Antarctic ice shelf thickness changes and basal melt rate estimates, and added additional scripts to the public `captoolkit` repository, which we believe will help the readers who wish to replicate our analysis (https://github.com/nasa-jpl/captoolkit/blob/master/RECIPE.md).

We can confidently say that this is the most comprehensive paper to date describing the estimation of ice shelf change/melting from satellite altimetry. We hope that you will find the changes satisfactory and that our work will be of interest and value to *The Cryosphere* readers.

Sincerely,

Fernando Paolo
and co-authors

---

## Author Response (AR2)

Dear Editor and Reviewers,

We would like to thank you and the reviewers once again for your constructive feedback on our manuscript. We have addressed **all** remaining requests from Reviewer #1 (see point-by-point below), and also explain below why we are unable to address Reviewer #2's suggestion. We hope this version of the manuscript is now sufficient for publication.

**Reviewer 1:**

*The revised manuscript has been largely improved by the authors and I only have technical comments on this new version:*

*L20: 'Our study' instead of 'or'*

> Thanks for catching this.

*L78: Please be more specific than 'ice change', i.e. 'ice and firn thickness change'*

> We now say "Measured height changes reflect the (solid) ice thickness change and the changes in air content within the firn layer"

*L82: 'those processes deemed most relevant to glacier studies', can you enumerate those relevant processes here?*

> We now say: "GEMB is run as a module of NASA's open-source Ice-sheet and Sea-level System Model (ISSM). It is a column model (no horizontal communication) of intermediate complexity, simulating thermal diffusion, shortwave subsurface penetration, meltwater retention, percolation and refreeze, effective snow grain size, dendricity, sphericity, and compaction."

*It seems that the revised figures have not been included in the revised manuscript? Please check that the figures presented are up to date.*

> We have updated the figures.

*Figure 5, right panel: The ice shelves are not labelled?*

> We have added labels for each ice shelf: Pine Island, Thwaites remnant, Thwaites calved, Crosson, and Dotson

*Figure 6: The black line is still on the inset map?*

We have removed the black line.

*Figure 9: This figure has not been changed, though the authors had agreed to change the size of the green vectors and add the date of the grounding line?*

We have increased the size of the vectors, changed their color to white for clarity, and added the following text to the caption of the figure: "grounding lines compiled from InSAR measurements taken from 1994 to 2009 (Rignot et al., 2011).

**Reviewer 2:**

*Dear Fernando Paolo and co-authors,*

*I think the revised manuscript is a nice re-worked version of the original manuscript which absolutely IMHO increased the readability. Moreover, I would like to thank the authors for their effort to make the code much more accessible. I personally think this is great for open science!*

*Since the authors did not do a comment-by-comment response to the reviews, I have however checked some of my original comments and repeated one important (rephrased) comment below. I do suggest that addressing this comment is potentially important, but I leave it at the editor's decision to decide if it is absolutely needed or not*

*Comment: the authors currently attribute the change in thinning rates to two main causes: i) changes in ice flow and ii) changes melt rates (as the remainder of flow+SMB). However, the relative contribution of each factor for each ice shelf remains unclear (e.g., it is shown for only four ice shelves in Table 2). I really do think it will help the reader to much better see the broader perspective of the drivers if the relative contributions of flow vs melt would be shown for all ice shelves. I would therefore strongly recommend to add the relative contributions for the ice flow and melt and the change in thinning rates for all ice shelves. This could be done for example as extra columns in Table 3 and as additional time series in the revised Figure 14.*

We agree that knowing the changes in ice flow of all ice shelves would be valuable additional information to know. Unfortunately, we do not have time-dependent

velocity data outside of the ice shelves presented in Table 3 (Pine Island, Thwaites, Crosson, and Dotson). In fact, we are not aware of existing velocity records for most ice shelves outside of the Amundsen Sea sector for the time period of our study. So we could not calculate the change in ice divergence for most ice shelves. We also note that Gardner et al. 2018 showed no significant changes in ice velocity for most East Antarctic ice shelves in more recent years, meaning that changes in ice divergence are likely a minor contribution to the overall thinning/thickening rate of those ice shelves.

Sincerely,

Fernando Paolo
and co-authors